# Spurious regulatory connections dictate the expression-fitness landscape of translation factors

Jean-Benoît Lalanne[1,2,†] (iD), Darren J Parker[1,‡] & Gene-Wei Li[1,*] (iD)

## Abstract

During steady-state cell growth, individual enzymatic fluxes can be directly inferred from growth rate by mass conservation, but the inverse problem remains unsolved. Perturbing the flux and expression of a single enzyme could have pleiotropic effects that may or may not dominate the impact on cell fitness. Here, we quantitatively dissect the molecular and global responses to varied expression of translation termination factors (peptide release factors, RFs) in the bacterium *Bacillus subtilis*. While endogenous RF expression maximizes proliferation, deviations in expression lead to unexpected distal regulatory responses that dictate fitness reduction. Molecularly, RF depletion causes expression imbalance at specific operons, which activates master regulators and detrimentally overrides the transcriptome. Through these spurious connections, RF abundances are thus entrenched by focal points within the regulatory network, in one case located at a single stop codon. Such regulatory entrenchment suggests that predictive bottom-up models of expression-fitness landscapes will require near-exhaustive characterization of parts.

**Keywords** expression-fitness landscape; multiscale measurements; peptide chain release factors; regulatory entrenchment; translation factors
**Subject Categories** Biotechnology & Synthetic Biology; Translation & Protein Quality
**Mol Syst Biol. (2021) 17: e10302**

## Introduction

Formulating predictive models connecting genetic information to phenotypes constitutes an overarching goal in genomics and systems biology (Ostrov *et al*, 2019; Shendure *et al*, 2019; Lopatkin & Collins, 2020). In single-celled microbes, the relationship between genotype and phenotype can be conceptually decomposed into two distinct maps: the first, relating genome sequence to gene expression and the second, connecting gene expression to whole-cell properties such as proliferation. Rapid progress on the characterization of *cis*-regulatory elements, spurred by integration of massively parallel reporter assays (Patwardhan *et al*, 2009, 2012; Sharon *et al*, 2012) with novel computational frameworks (Rosenberg *et al*, 2015; Jaganathan *et al*, 2019; Bogard *et al*, 2019; de Boer *et al*, 2020), has achieved headway in predicting expression features from DNA sequence (Cambray *et al*, 2018; Sample *et al*, 2019; Agarwal & Shendure, 2020; preprint: Urtecho *et al*, 2020). By contrast, expression-fitness landscapes, defined as the distinct relationships between the expression level of individual genes and the cell fitness, remain understudied despite being the basis of selective pressures on protein abundances. This information gap limits both the engineering of complex biological functions and the interpretability of genetic variation.

Predicting the shape of expression-fitness landscapes requires quantitative characterization of the cellular state at multiple levels. Although numerous expression-fitness landscapes have been previously mapped, e.g., (Tubulekas & Hughes, 1993; Dekel & Alon, 2005; Chou *et al*, 2014; Li *et al*, 2014; Keren *et al*, 2016; Knöppel *et al*, 2016; Duveau *et al*, 2017; Palmer *et al*, 2018; Schober *et al*, 2019; Hawkins *et al*, 2020; Jost *et al*, 2020; Kavčič *et al*, 2020; Parker *et al*, 2020; preprint: Arita *et al*, 2021; Mathis *et al*, 2021), these measurements rarely include concomitant assessment of the internal cell state following perturbations (but see Jost *et al*, 2020; Parker *et al*, 2020). With limited information bridging the molecular to cellular scales, the root causes of observed fitness defects are challenging to pinpoint (Fig 1A). In particular, changes in enzyme levels not only directly affect flux and growth (Fig 1A, inset i for the case of translation factors) (Ehrenberg & Kurland, 1984; Klumpp *et al*, 2013; Li *et al*, 2014), but can also have indirect pleiotropic effects that take the form of damage propagation across three connected levels of biological organization (Fig 1A, inset ii). First, a reduced enzymatic flux could affect the expression of other genes via specific molecular mechanisms (mechanistic level). Second, these proximal changes in expression could ripple through the regulatory network, leading to further changes in expression genome-wide (regulatory level). Third, each terminal downstream expression change could have an impact on fitness (systemic level). Whether selective pressures on the abundance of proteins predominantly relate to direct impacts on flux

1 Department of Biology, Massachusetts Institute of Technology, Cambridge, MA, USA
2 Department of Physics, Massachusetts Institute of Technology, Cambridge, MA, USA
*Corresponding author. Tel: +1 617 324 6703; E-mail: gwli@mit.edu
†Present address: Department of Genome Sciences, University of Washington, Seattle, WA, USA
‡Present address: Biosciences Division, Oak Ridge National Laboratory, Oak Ridge, TN, USA

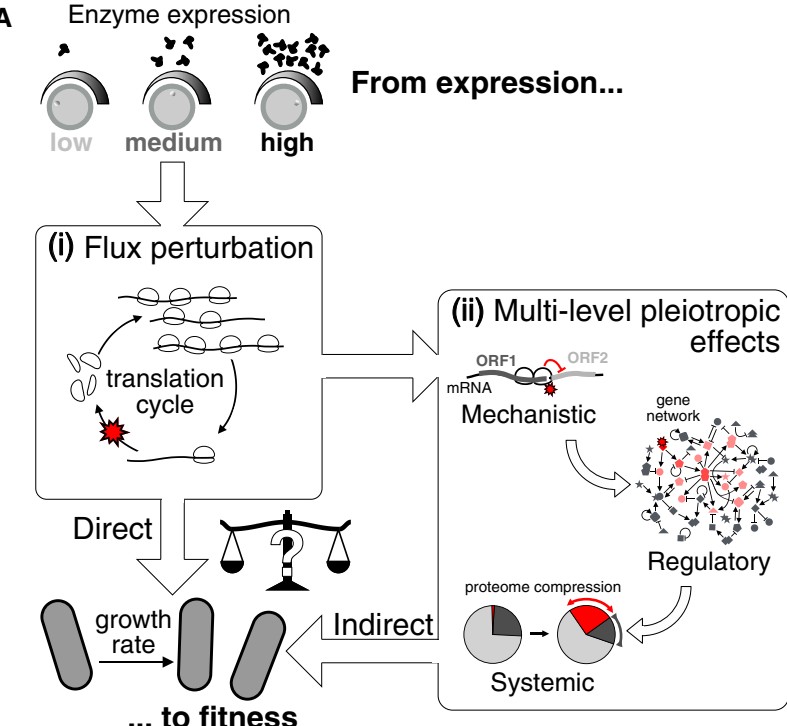

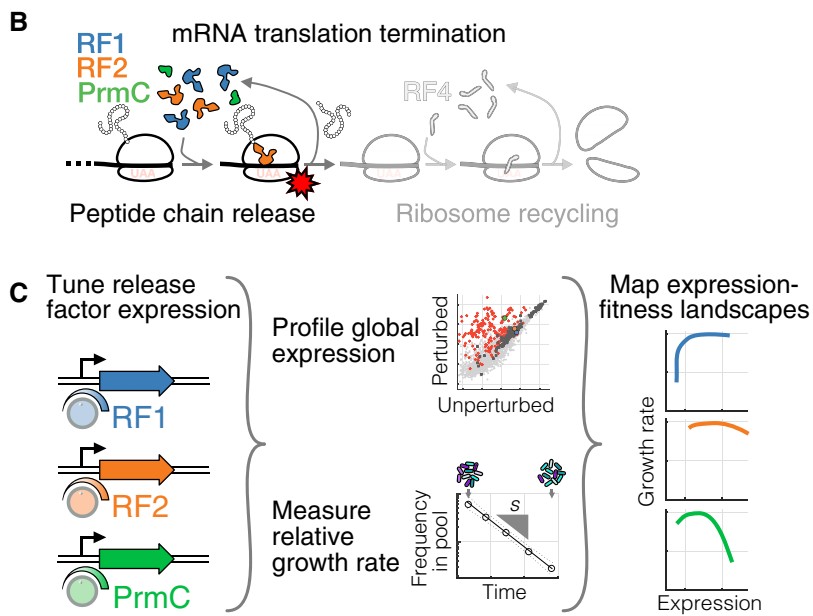

**Figure 1. Mapping the underlying determinants of the release factor expression-fitness landscapes.**

A   Expression-fitness landscapes, which connect enzyme expression (microscopic variable) to the growth rate (cellular phenotype), can be dictated by direct or indirect effects. Inset (i): direct effects correspond to reduction in the flux cognate to the perturbed enzyme (protein synthesis rate in the case of translation factors). Inset (ii): indirect effects result from pleiotropic propagation across mechanistic, regulatory, and systemic levels.

B   As a case study, the expression of peptide chain release factors (RFs: RF1, RF2, and associated methyltransferase PrmC), involved in the first step of mRNA translation termination, was tuned around endogenous levels.

C   Strains with inducible copies of RFs, and deleted endogenous genes, were used to systematically vary RF expression. The resulting impacts on the cell internal state (RNA-seq, ribosome profiling) and relative growth rate $s$ (competition experiments) were measured, leading to precise mapping of expression-fitness landscapes.

Data information: See also Figs EV1 and EV2 for details on strains and measurement platform.

or are rather dominated by indirect cellular responses remains unresolved.

Here as a case study, we systematically vary the expression of enzymes involved in the core process of mRNA translation termination (Fig 1B) in Gram-positive bacterium *Bacillus subtilis*. We focus on peptide chain release factors, RF1, RF2, and their methyltransferase PrmC (hereafter collectively referred to as release factors, RFs). RF1 and RF2 catalyze the first step of translation termination (Bertram *et al,* 2001), recognizing stop codons and releasing completed peptides from the ribosome (Fig 1B). RF1 and RF2 directly interact with the ribosome (Scolnick *et al,* 1968; Petry *et al,* 2005) and have partially overlapping specificities (Scolnick *et al,* 1968) (RF1 recognizes stops UAA/UAG, and RF2 stops UAA/UGA). PrmC post-translationally modifies RF1 and RF2, thereby increasing their catalytic activity (Heurgué-Hamard *et al,* 2002; Nakahigashi *et al,* 2002; Mora *et al,* 2007). We targeted translation termination because the resulting downstream changes in expression were anticipated to be modest: given that translation is initiation-limited on most mRNAs (Laursen & Sørensen, 2005), mildly decreasing termination rate was expected to reduce global ribosome availability without altering protein production on a gene-by-gene basis. A better understanding of the physiology of translation termination stress has relevance in synthetic biology, for example in the context of genome-wide stop codon reassignment (Johnson *et al,*2011, 2012; Lajoie *et al,* 2013; Wannier *et al,* 2018; Fredens *et al,* 2019).

Through precise measurements of transcriptomes, global translational responses, and cell fitness (defined here as the population growth rate in exponential phase), we elucidate the underlying determinants of the RF expression-fitness landscapes (Fig 1C). We find that idiosyncratic and indirect inductions of regulatory programs are associated with decreases in growth rate in multiple directions of the RF expression subspace. We term such situation "regulatory entrenchment", whereby the fitness defect caused by perturbing a protein's expression is strongly and spuriously aggravated by the gene regulatory network. Further, we reconstruct links that connect the initial microscopic perturbation to system-wide effects. At the mechanistic level, we find that occlusion of ribosome-binding sites during RF depletion is a common phenomenon leading to changes in expression stoichiometries between adjacent, co-transcribed genes. In particular, we identify the stop codon of a single regulator as a molecular Achilles heel, sensitizing the entire cell to specific RF perturbations. At the regulatory level, we show that removing one gene can mute pleiotropic changes and liberate RF from regulatory entrenchment. Finally, at the systemic level, we report passive proteome compression as a quantitatively tractable cause of growth defects upon massive activation of a regulon.

Our multiscale characterization provides a concrete example of how distal yet focal events triggered by targeted molecular perturbations can have system-wide impacts. Here, the cells' susceptibility to expression perturbations of specific enzymes is not simply related to the magnitude of the changes in the associated cognate flux, but is instead dictated and amplified by sensitive nodes in the regulatory network. These results underscore the viewpoint that a quantitative understanding of the full system (Karr *et al,* 2012; Boyle *et al,* 2017; Liu *et al,* 2019) might be necessary to predict even qualitatively the shape of expression-fitness landscapes.

# Results

## Linking RF perturbations to changes in genome-wide expression and fitness

We used a combination of genetic tools and high-throughput measurements to map RF expression-fitness landscapes, as well as the underlying mechanistic, regulatory, and systemic responses. We created strains in which two of the three factors (PrmC, RF1, and RF2) can be tuned orthogonally (Fig EV1E and F): PrmC and RF1 in one strain, and PrmC and RF2 in another (with the autoregulatory frameshift removed for RF2 (Craigen *et al,* 1985; Craigen & Caskey, 1986), Fig EV1A–C). The range of these tunable expression systems spanned 31- to 111-fold and was centered near their respective endogenous levels (Fig EV1D). The global gene expression changes resulting from these perturbations were probed with RNA-seq (Parker *et al,* 2019) (Materials and Methods), with highly reproducible approaches (Appendix Fig S1). Expression levels were converted to fractions of the proteome using ribosome profiling (Ingolia *et al,* 2009; Li *et al,* 2014) (Materials and Methods for details and assumptions). Ribosome profiling further provided a high-resolution view of translation state in a subset of conditions. In order to precisely measure fitness, defined here as relative population growth rate in exponential phase, we designed a DNA-barcoded competition assay (Smith *et al,* 2009; Parker *et al,* 2020) with ± 1% precision (± 2$\sigma$, Fig EV2, Materials and Methods). This combination of targeted proteome perturbation with precision transcriptomic and fitness measurements enabled a multiscale assessment of the RF expression-fitness landscapes.

## Perturbing the expression of different RFs decreases fitness through distinct physiological routes

Using our measurement platform, we directly confirmed that endogenous RF expression in exponential phase maximizes growth rate. For conditions at or near endogenous levels of RF1, RF2, and PrmC (dashed lines, Fig 2E–G), the fitness is indistinguishable from wild-type strains (shaded gray area in Fig 2E–H corresponds to experimental precision in fitness, $|s| < 2\sigma_s = 1.2\%$, Materials and Methods). On the other hand, perturbing RF expression away from endogenous level led to growth defects in most directions of the RF subspace (Fig 2E–H). Only RF1 overexpression did not cause a measurable decrease in fitness, in part because of the limited maximal achievable level for this particular inducible construct ($\approx 3\times$, Fig EV1D). These results suggest that RF expression is optimized for exponential phase growth, akin to the expression of several other factors (Ehrenberg & Kurland, 1984; Dekel & Alon, 2005; Li *et al,* 2014; Parker *et al,* 2020), and is consistent with the tight evolutionary conservation of expression stoichiometry in biological pathways (Lalanne *et al,* 2018).

Away from the optimum, distinct RFs displayed expression-fitness landscapes of different shapes. For example, PrmC has a much more severe growth defect than RF2 at the same levels of overexpression (Fig 2F vs. 2G), and RF1 knockdown leads to a near-vertical drop-in fitness (Fig 2E), which is more severe than RF2 or PrmC knockdown (though our expression constructs had higher baseline expression for the latter two). Beyond these qualitative differences, our global expression quantification in all these conditions provides

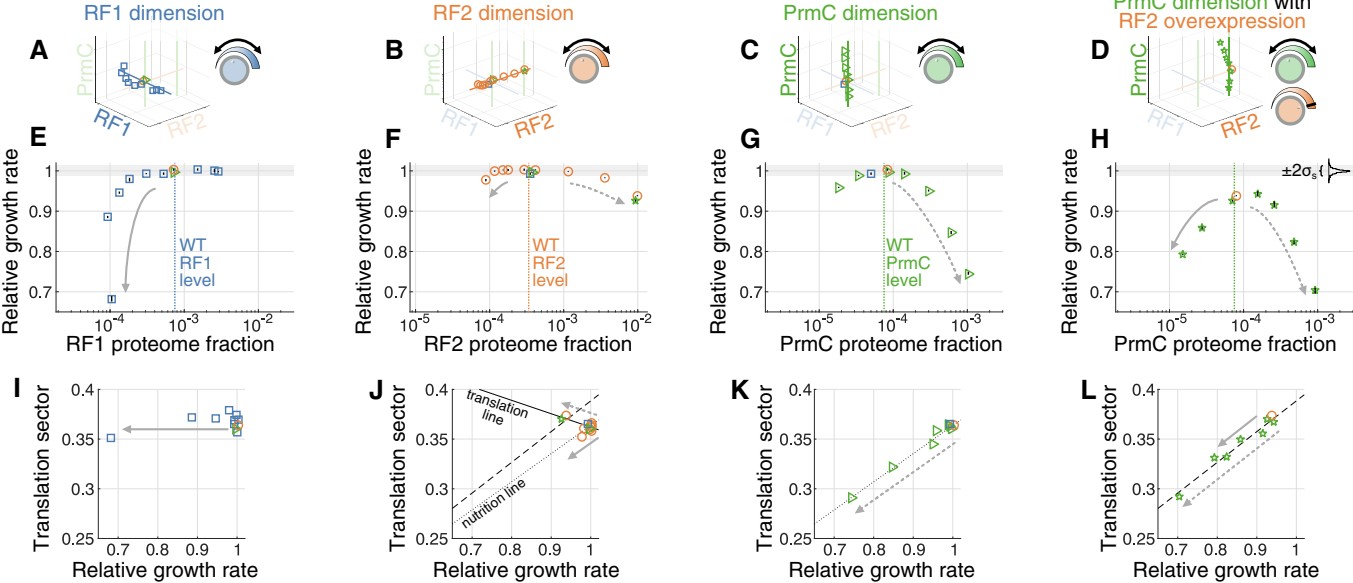

**Figure 2. Diverse fitness landscapes and physiological trajectories upon RF expression perturbation.**

A–D    Profiled orthogonal directions of the RF expression subspace, respectively, scanning along the dimension of (A) RF1, (B) RF2, (C) PrmC, and (D) PrmC with RF2 overexpression. Axes correspond to expression levels of RF1, RF2, and PrmC.

E–H    Cell exponential growth rate s measured by competition (relative to wild-type) at corresponding RF levels (reported in units of proteome fraction, derived from a ribosome profiling calibration, Materials and Methods) shown in (A–D). Endogenous levels of RFs are indicated with dashed vertical lines. Gray shadings mark the precision of our fitness measurement, defined as $\pm 2\sigma_s = \pm 1.2\%$, where $\sigma_s$ is the standard deviation in the measured relative growth rate among isogenic redundantly barcoded strain (see Fig EV2E), with the distribution shown as inset in panel (H). Relative fitness value reported corresponds to the median across isogenic barcoded pairs, and vertical black bars delineate 25th to 75th percentile among such pairs (between 21 and 28 isogenic pairs, see Dataset EV6) for a single experiment and is typically smaller than the plot symbol.

I–L    Trajectories following RF perturbation in the space of relative growth rate vs. estimated proteome fraction to translation proteins (translation sector).

Data information: Matched arrows across panels (E–H) and (I–L) show direction of increasing perturbations in the fitness landscape. Lines in (J–L) correspond to least-square fits. Lines in panels (K and L) have been reproduced in panel (J) to highlight of additivity of trajectories under PrmC perturbation with RF2 overexpression. See also Figs EV1–EV4.

a way to assess whether RF-specific fitness defects nevertheless correspond to similar underlying physiological states generic to translation termination defects.

Projecting complex phenotypic data onto low-dimensional manifolds can provide insight on the physiological state underlying growth defects (Scott *et al,* 2010, 2014; You *et al,* 2013; Hui *et al,* 2015). Two cellular-level quantities are of particular interest: (i) the growth rate $\lambda$ and (ii) the total expression of the mRNA translation machinery, quantified as the summed proteome fraction of translation proteins, termed the translation sector, $\phi_R$. Hwa *et al* have shown (Scott *et al,* 2010; You *et al,* 2013; Zhu *et al,* 2016, 2019) that trajectories in the space of growth $\lambda$ vs. translation sector $\phi_R$ upon a series of increasingly severe perturbations are intimately related to global regulation (Appendix Supplementary Methods, Appendix Fig S2). In *Escherichia coli,* they found that under the numerous ways to inhibit translation, the translation sector $\phi_R$ increases concomitantly with a decrease in the growth rate (along the "translation line", Appendix Fig S2) as a compensation mechanism (Scott *et al,* 2010). By contrast, decreasing the nutritional quality of the medium leads to a decrease in growth paralleled by a decrease in the translation sector $\phi_R$ (along the "nutrition line", Appendix Fig S2), a long-known property in bacterial physiology (Schaechter *et al,* 1958). Under the bacterial growth laws established in *E. coli,* the states of

cells with RF-specific fitness defects were anticipated to collapse on the translation line. In effect, perturbing RF expression was expected to reduce the translation termination rate and thus to reduce ribosome availability.

Surprisingly, the different RF expression perturbations displayed diverse physiological trajectories (Fig 2I–L). The sharp drop in growth under RF1 knockdown led to little change in the translation sector $\phi_R$ (Fig 2I), analogously to the trajectories associated with non-translation-targeting antibiotics (Scott *et al,* 2010). By contrast, the growth defects upon PrmC expression perturbations were associated with a decrease in the translation sector (Fig 2K and L), leading to movement along the nutrition line, which was unexpected given the fixed quality of the growth medium. RF2-perturbed cells moved slightly in different directions following overexpression (translation line) and knockdown (nutrition line, although high basal activity of our expression construct limited the magnitude of accessible growth defects; Fig 2J). Interestingly, physiological changes were additive under combined RF perturbations: tuning PrmC expression in combination with RF2 overexpression led to movement along the nutrition line, but shifted by the movement along the translation line by the RF2 perturbation (dashed vs. dotted lines in Fig 2J–L, Appendix Supplementary Methods).

The physiological divergences observed for the different RFs perturbations were intriguing given the similar involvement of these three proteins in translation termination and pointed to possible RF-specific pleiotropic effects. Analyzing the transcriptomes following perturbations revealed distinct responses along the orthogonal RF expression directions. PrmC perturbations massively induced the general stress $\sigma^B$ regulon (Figs 3A and B, and EV3A–D), RF2 perturbation led to little changes across the full range of expression, except for a modest $\sigma^B$ induction at maximal knockdown (Fig EV3O and P). RF1 knockdown led to changes in motility and biofilm genes (Fig EV4C and D). Together, these results suggest that each RF has a mechanistically distinct relationship between expression and fitness.

### Fitness defects during PrmC overexpression can be traced back to regulatory response by $\sigma^B$ activation

To further identify the origin of some of the observed growth defects, we focused on perturbations to PrmC. The growth defect following PrmC overexpression (Fig 2G) coincided with dramatic activation of the general stress regulon, which is driven by the alternative sigma factor $\sigma^B$ (gene *sigB*) (Helmann *et al*, 2001; Petersohn *et al*, 2001; Price *et al*, 2001; Price, 2002; Hecker *et al*, 2007; Zhu & Stülke, 2018), from 2% basal proteome synthesis fraction in wild-type to > 20% (Fig 3A and C, see Materials and Methods for ribosome profiling calibration). This increase in the expression of $\sigma^B$ genes was accompanied by a corresponding decrease in the translation sector $\phi_R$ (Fig 3D and E), which raised the possibility that a substantial portion of the growth defect following PrmC overexpression was associated with the physiological burden of expressing > 100 $\sigma^B$-dependent genes at high levels (Benson & Haldenwang, 1992; Boylan *et al*, 1992; Bernhardt *et al*, 1997). Alternatively, induction of the stress regulon might have directly responded to and played a role in mitigating, translational stress resulting from PrmC expression changes. As additional evidence of systemic stress, PrmC overexpression leads to global changes in gene expression beyond the $\sigma^B$ regulon (broadening of fold-change distribution, Fig 3A inset). Whether these global changes were the result of the PrmC perturbation, or secondary to $\sigma^B$ activation, was to be clarified.

To determine whether $\sigma^B$ exacerbates or mitigates fitness defects under PrmC perturbations, we profiled RF-inducible cells lacking the *sigB* gene ($\Delta sigB$). As expected, deleting *sigB* completely abrogated regulon induction (Fig 3A vs. B and C, dashed red arrows), but surprisingly also restored genome-wide expression to near endogenous levels (Fig 3B inset). Hence, global expression changes (gray line in inset Fig 3A) upon PrmC overexpression were caused by $\sigma^B$ activation and its downstream consequences (e.g., sigma factors competition (Farewell *et al*, 1998)). Importantly, the PrmC overexpression growth defect was completely rescued following *sigB* deletion (Fig 3G dashed red arrow), concomitantly with the restoration of the translation sector to endogenous levels (Fig 3D and E dashed red arrows). These observations were reproducible under simultaneous PrmC/RF2 overexpression (Fig EV3A–I, Appendix Supplementary Methods).

Taken together, these results suggest that the gene *sigB* constitutes a *trans*-modifier (Hou *et al*, 2019) of the PrmC expression-fitness landscape. Through its activation, $\sigma^B$ drastically exacerbates

the cell's sensitivity to expression perturbation of distal gene *prmC*. Overall, the induction of the general stress regulon has no measured benefit in the context of PrmC perturbations. Gene *sigB* therefore "entrenches" PrmC expression by imposing a massive physiological burden to cells that is in addition to, and much larger than, the effect directly related to perturbing PrmC.

### Systemic proteome compression by $\sigma^B$ explains fitness defect

We hypothesized that the growth defect resulting from $\sigma^B$ activation mainly arose from the cost of expression of regulon genes. The burden of expressing proteins has been extensively characterized in *E. coli* (Andrews & Hegeman, 1976; Dong *et al*, 1995; Stoebel *et al*, 2008; Scott *et al*, 2010). Gratuitous expression of genes causes "proteome compression", whereby the abundances of other proteins globally decrease in response to the finite total synthesis capacity of the cell (Scott *et al*, 2010). The resulting reduction in the translation sector, responsible for generating biomass, leads to a growth defect, which would be in addition to any protein-specific effects such as misfolding or aggregation. Bacterial growth laws, established in *E. coli*, predict the quantitative relationship between proteome fraction of gratuitous proteins and growth rate under this proteome compression model (Appendix Supplementary Methods).

Despite the complexity of the induced regulon (> 100 diverse genes) and the resulting secondary changes in gene expression (Fig 3A), $\sigma^B$ activation did lead to simple compression of the translation machinery (Fig 3E and F). Indeed, the translation sector (full line, Figs 3E and EV3G) and growth rate (full line, Figs 3H and EV3I) linearly decreased in a one-to-one proportion with the increase in excess proteome fraction to the $\sigma^B$ regulon. Interestingly, we observed a smaller growth defect than expected from the *E. coli* growth law (dashed vs. full lines, Figs 3E and H, and EV3G and I, Appendix Supplementary Methods). Further assessment of the generality of growth laws to species other than *E. coli* will constitute promising future research avenues.

Compression of the translation sector due to $\sigma^B$ activation provides a systemic explanation for the movement along the nutritional line under PrmC perturbation (Fig 2K and L). Abrogation of this effect following *sigB* deletion (Figs 3D and E, and EV3F and G, dashed arrow) highlights that physiological trajectories can be modulated by individual regulatory proteins, as has been also demonstrated for tRNA synthetases and the stringent response in *B. subtilis* (Parker *et al*, 2020). However, unlike the stringent response, which is cognate to synthetase perturbation and beneficial to cell growth, the $\sigma^B$ response here is detrimental.

### Activation of regulator $\sigma^B$ does not result from general stress-sensing mechanisms

The detrimental effect on fitness of $\sigma^B$ activation under RF stress (Fig 3G) suggests that its induction is idiosyncratic in this context, as opposed to being the result of a cognate sensing mechanism. We hypothesized that RF-specific perturbations led to changes in a small number of proteins that ultimately activated $\sigma^B$. In support of this view, $\sigma^B$ is only activated when certain RFs are perturbed: Varying RF1 expression over the full achievable range (0.1× to 3× endogenous) did not change $\sigma^B$ regulon expression (Fig EV4C), whereas RF2 knockdown, but not overexpression, was associated with $\sigma^B$

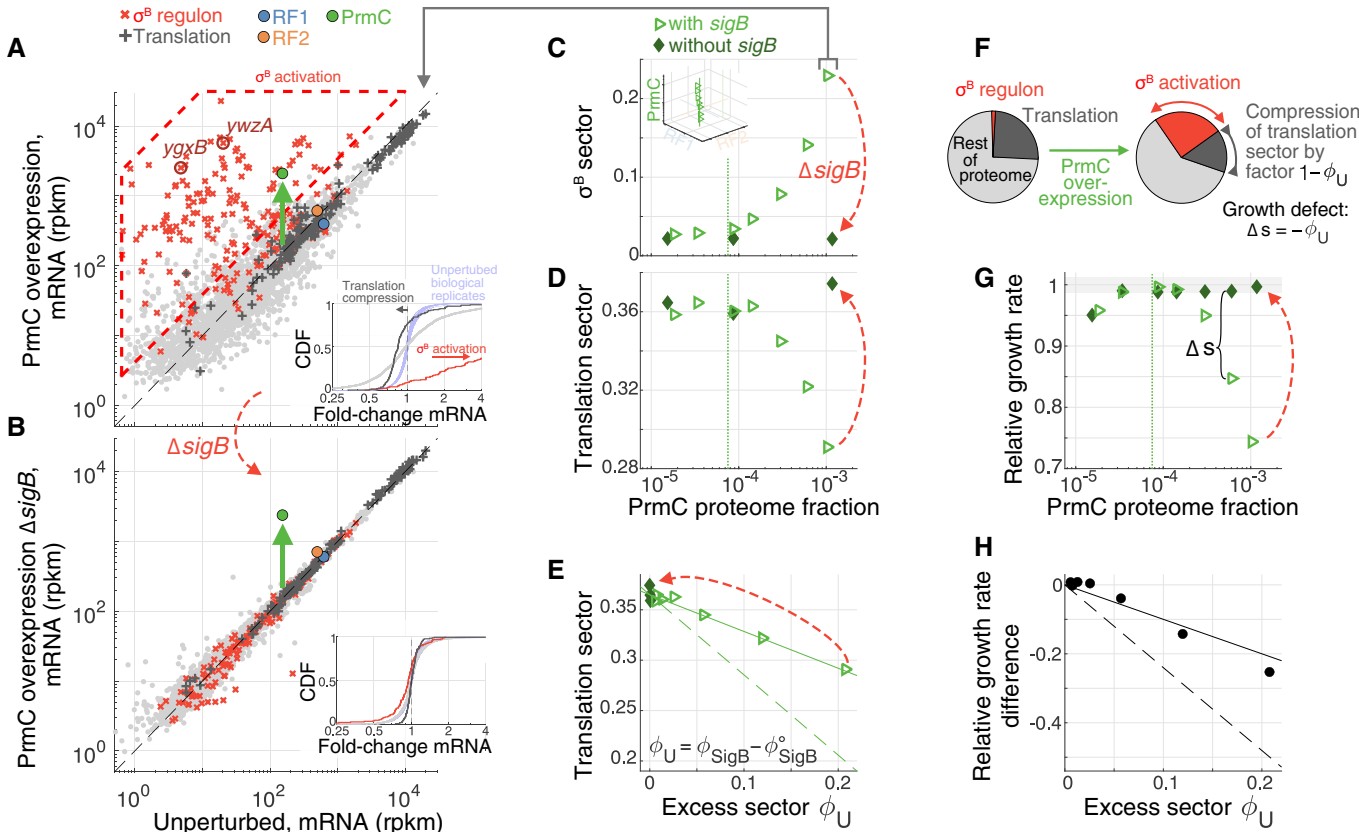

**Figure 3. σ B regulon induction upon PrmC overexpression compresses the translation sector.**

A   mRNA levels (reads per million mapped reads per kilobase, rpkm, genes with > 5 reads mapped shown) for maximal PrmC overexpression (green arrow) versus unperturbed cells (average across control datasets, Materials and Methods). σ B regulon members and translation-related proteins are marked in red × and dark gray +, respectively. Targets for RT–qPCR measurements of σ B induction (*ygxB*, *ywzA*, Fig 5) are highlighted in dark red. mRNA levels for RF1 (blue), RF2 (orange), and PrmC (green) are marked by dots (corrected for translation efficiency of ectopic expression constructs, Materials and Methods). σ B regulon activation is marked by dashed red polygon. Inset shows cumulative distribution of fold-changes in mRNA levels (red σ B regulon, dark gray translation, pale gray rest of proteins). σ B activation and translation compression are highlighted by arrows indicating shift in median expression. As a comparison, distribution of fold-changes for all genes among unperturbed replicates are show in light blue.
B   Same as (A), but with a strain harboring a deletion of gene *sigB*, which abrogates σ B regulon activation, and restores genome-wide expression levels despite PrmC overexpression (light gray line in inset).
C   Quantification of the proteome fraction to the σ B regulon (σ B sector) as a function of PrmC (see Materials and Methods for calibration from transcriptome to proteome fraction). Dashed vertical line marks endogenous PrmC level. Inset reproduces broader context of data in RF expression subspace (Fig 2C).
D   Similar to (C), but quantifying proteome fraction to the translation sector.
E   Proteome fraction of the translation sector as a function of the excess proteome fraction to the σ B regulon, denoted $\phi_U$. Dashed line corresponds to growth laws prediction (Appendix Supplementary Methods, equation 1, using parameters $\kappa_n$, $\kappa_t$, $\phi_o$, and $\phi_R^{max}$ obtained from fits in Fig 2J and K), full line corresponds to decrease by factor $1 - \phi_U$.
F   Schematic illustration of passive proteome fraction compression under σ B activation (increase in regulon expression).
G   Relative growth rate as a function of PrmC level, with and without *sigB*.
H   Growth difference with and without *sigB* (corresponding to Δs panel G), as a function of excess proteome fraction to the σ B regulon. Dashed lines are growth law prediction (Appendix Supplementary Methods, equation 2), full line corresponds to $-\phi_U$.

Data information: In panels (C–E and G), open light green triangles correspond to cells with *sigB*, and filled dark green diamonds to cells without *sigB* (deletion). See also Figs EV3 and EV4.

activation (Fig EV3O and P, and lowering basal RF2 expression further increased σ B activation, see Dataset EV8). Similar to PrmC overexpression, the fitness defects under RF2 knockdown were rescued by *sigB* deletion (Fig EV3Q), also arguing for regulatory entrenchment of RF2 expression by σ B.

To confirm that σ B activation arose from perturbations specific to distinct RFs and not from general translation stress-sensing mechanisms, we used ribosome profiling to quantify translation in cells

acutely depleted of RFs. We used CRISPRi transcriptional interference (Qi *et al*, 2013; Peters *et al*, 2016) to separately knockdown RF1/PrmC (co-transcribed) and RF2. In both cases, we found evidence of translational stress in the forms of idle ribosomes at the corresponding stop codons and queuing upstream (Fig 4A), similar to what was previously observed under different translation termination stresses (Baggett *et al*, 2017; Mangano *et al*, 2020; Saito *et al*, 2020). Specifically, we observed queues at UAG stops under

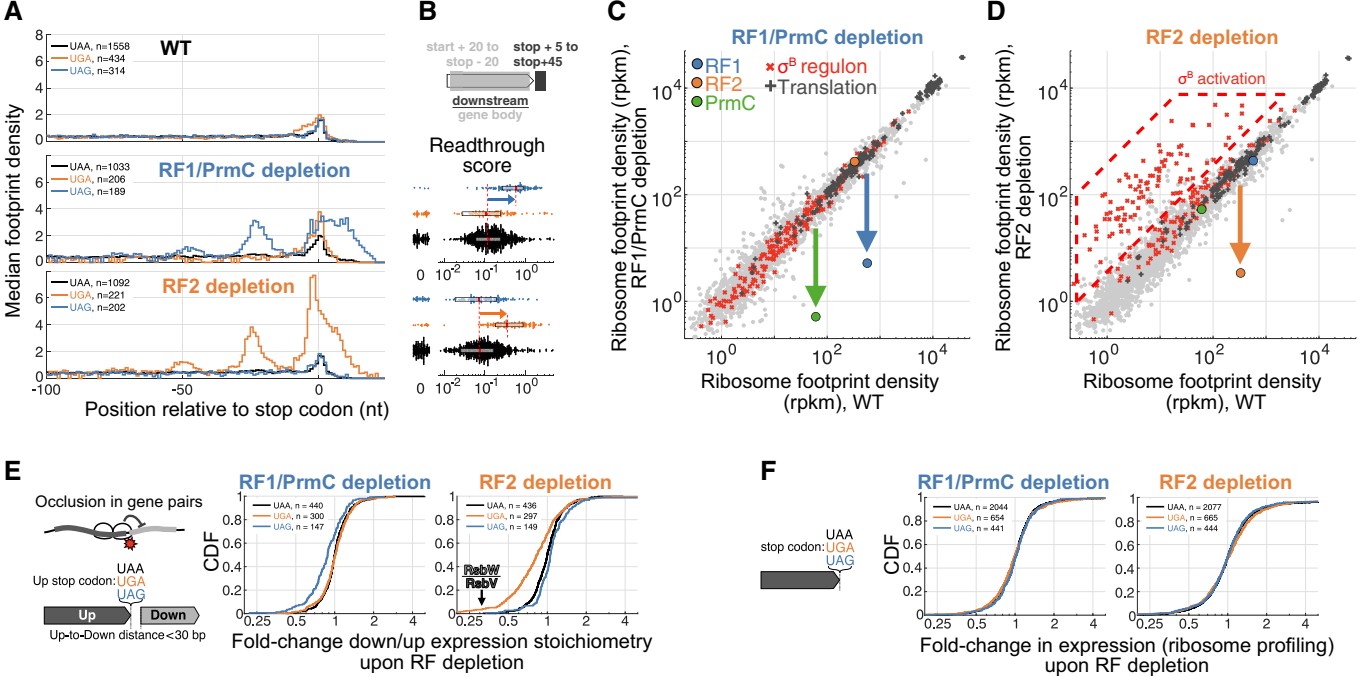

**Figure 4. Translation termination stress and gene expression changes upon acute RF depletion with CRISPRi.**

A  Metagene trace (Materials and Methods) of gene-normalized ribosome footprint read density (center-mapped, genes with footprint density > 0.5/nt) upstream of stop codons stratified by stop codon. Wild-type (top), RF1/PrmC depletion (middle), and RF2 depletion (bottom) are shown. Ribosome queues upstream of stop codon cognate to RF perturbations can be seen.

B  Translation readthrough score for isolated genes (Materials and Methods) under acute R1/PrmC and RF2 CRISPRi knockdown. Points in beeswarm plot correspond to individual genes, overlaid box plot highlighting the interquartile range (25th to 75th percentile, median red mark). Fivefold increase in translational readthrough is seen for genes terminating with the stop codon cognate to the knocked-down RF (arrows).

C, D  Comparison of expression (ribosome footprint density, rpkm) for (C) RF1/PrmC depletion, and (D) RF2 depletion, vs. wild-type. Regulons and RF are marked as in Fig 3A. RF2 depletion leads to $\sigma^B$ activation, in contrast to RF1/PrmC depletion.

E  Fold-change in the downstream-to-upstream expression (ribosome profiling) stoichiometry upon RF depletion for co-directional genes within 30 bp, and stratified by the stop codon of the upstream gene. A mild but systematic and stop codon-specific decrease in downstream gene expression is seen (RF1/PrmC: median UAG fold-change compared with UAA ($FC_{UAG}$) = 0.88, $P < 10^{-6}$; RF2: $FC_{UGA}$ = 0.82, $P < 10^{-6}$, $P$-values from stop codon reshufflings, Materials and Methods). We ascribe this effect to obstruction of downstream ribosome initiation by idle upstream terminating ribosomes. Some protein pairs, such as RsbW/RsbV, are affected more strongly.

F  Fold-change in expression between RF depletion and wild-type stratified by stop codon of genes. No systematic effect for the different stop codons is seen (RF1/PrmC: $FC_{UAG}$ = 1.01, $P$ = 0.69; RF2: $FC_{UGA}$ = 1.02, $P$ = 0.86, $P$-values from stop codon reshufflings, Materials and Methods), indicating lack of strong change in expression for genes with compromised translation termination.

Data information: See also Figs EV3 and EV5, Appendix Figs S3 and S4.

RF1/PrmC depletion, queues at UGA stops under RF2 depletion, and no queues in wild-type, or at UAA stops in either depletion. Queues were longer on mRNAs with high translation efficiency, as expected from models of stochastic queuing (Bergmann & Lodish, 1979; Mitarai *et al*, 2008; Baggett *et al*, 2017; Saito *et al*, 2020) (Fig EV5B, C and H, Appendix Supplementary Methods). The stop codon-specific queuing is associated with translational readthrough downstream (Fig 4B), consistent with a recent observation based on inhibition of peptide release by the antimicrobial apidaecin in *E. coli* (Mangano *et al*, 2020). We also observed a trend of tetranucleotide-dependent (UGAN) readthrough for RF2 knockdowns (Materials and Methods, Appendix Fig S3) consistent with previous characterizations (Poole *et al*, 1995). The severity of the translation stress, as assessed by the magnitude of accumulation of ribosomes on stop codons and translational readthrough, was similar between the PrmC/RF1 and RF2 depletions (Fig 4A and B). Importantly however, a robust $\sigma^B$

activation was only observed under RF2, and not RF1/PrmC, knockdown (Fig 4C and D). The simultaneous presence of ribosome queues (Fig 4A, middle row) and absence of $\sigma^B$ activation under RF1/PrmC depletion (Fig 4C) confirmed that $\sigma^B$ induction was not channeled through a sensor of translation stress agnostic to the identity of ribosome-jammed mRNAs. $\sigma^B$ induction was instead likely due to gene-specific expression changes driven by RF stop codon specificities.

## RF depletion mechanistically leads to stoichiometric imbalance of co-transcribed genes

We hypothesized that a consequence of RF depletion is that the ribosomes idling at the corresponding stop codons would sterically occlude translation initiation of downstream genes. Because bacterial genes are often closely spaced in polycistronic operons,

elevated ribosome occupancy at stop codons may prevent other ribosomes from binding to the next gene (Fig 4E). Consistent with this hypothesis, we found that among co-directional genes separated by a ribosome footprint or less ($\leq$ 30 nt), the downstream genes exhibit a mild but stop-specific decrease in expression when the cognate RF is depleted (Fig 4E, RF1/PrmC: $FC_{UAG} = 0.88$, $P < 10^{-6}$; RF2: $FC_{UGA} = 0.82$, $P < 10^{-6}$, see Fig EV5D–F for control groups). This result differs from a lack of effects reported in *E. coli* during knockdown of ribosome recycling factors (RF4), which act on ribosomes post-RF1/RF2-mediated peptide release. These ribosomes may have different mobility and re-initiation properties (Saito *et al,* 2020). Although the effect size was modest overall, some gene pairs were affected more heavily. For example, highly translated upstream genes are correlated with stronger decrease in downstream expression Fig EV5I. Gene pairs joined by AUGA, i.e., the most common overlapping start and stop punctuation (Fig EV5J) possibly involved in translational coupling, did not show a substantial difference (Fig EV5K). Overall, these genome-wide measurements point to occlusion of ribosome-binding sites by upstream termination-idle ribosomes as a cause of RF-specific gene expression changes.

In contrast to the effect on co-transcribed genes downstream, the expression of genes ending with the compromised stop codons were not substantially affected either at the level of translation (Fig 4F, RF1/PrmC: median UAG fold-change compared with UAA ($FC_{UAG}$) = 1.01, $P = 0.69$; RF2: $FC_{UGA} = 1.02$, $P = 0.86$; $P$-values from stop codon reshufflings, Materials and Methods), or mRNA (Fig EV5G, RF1/PrmC: $FC_{UAG} = 0.98$, $P = 0.06$; RF2: $FC_{UGA} = 0.98$, $P = 0.02$). This lack of systematic changes in expression for genes with translation termination defects is consistent with the observed ribosome queues being too short to interfere with translation initiation at the upstream start of the ORF (queue size of < 4 stalled ribosomes globally, Fig 4A) (Bergmann & Lodish, 1979; Jin *et al,* 2002). This result also suggests that translational quality control and surveillance mechanisms, such as mRNA cleavage at empty A-site (Hayes & Sauer, 2003; Li *et al,* 2007) and ribosome collision sensors (Ferrin & Subramaniam, 2017), play a minor role in comparison with start codon occlusion for downstream genes in the current context.

### Mechanism for $\sigma^B$ activation by hypersensitivity of a single *cis*-element

Expression stoichiometry of genes *rsbW/rsbV*, whose ORFs overlap via the RF2 stop codon (−4 nt overlap AUGA, Figs 5B and EV5J), was one of the most sensitive to RF2 knockdown (3.0× change, Fig 4D, compared with 1.1× in RF1/PrmC knockdown). Intriguingly, these two proteins form the regulatory core of $\sigma^B$ activation. Upstream signaling events converge on a phosphorylation-dependent partner-switching mechanism involving the anti-sigma factor RsbW and the anti-anti-sigma factor RsbV, which, respectively, inhibits and activates $\sigma^B$ (Pané-farré *et al,* 2017). As additional control points, RsbW can deactivate RsbV by direct phosphorylation (Price, 2002; Hecker *et al,* 2007) and the three genes *rsbW*, *rsbV*, and *sigB* are co-transcribed under a $\sigma^B$-dependent promoter (among other transcript isoforms; Fig 5A). These interlocked positive and negative feedbacks (Fig 5A) render $\sigma^B$ activation hypersensitive to the stoichiometry of its regulators RsbV and RsbW (Igoshin *et al,* 2007; Locke *et al,* 2011).

Given that we observed $\sigma^B$ activation in RF2, but not RF1/PrmC, knockdown (Fig 4D vs. 4C), and that the expression of $\sigma^B$ regulators RsbV/RsbW was affected in these conditions (Fig 4D), we hypothesized that termination defects at the *rsbV* UGA (RF2-specific) stop codon were the cause of the $\sigma^B$ activation following RF perturbations. As such, switching the *rsbV* stop codon was predicted to be sufficient to rewire the susceptibility of $\sigma^B$ activation to different RF perturbation. For example, with a switched UAG stop for gene *rsbV*, $\sigma^B$ might be activated upon RF1, but not RF2, knockdown.

To test whether $\sigma^B$ activation sensitivity was indeed focally dependent on this *cis*-element, we generated a set of *rsbV* stop codon variants (UAA: RF1 and RF2 activity, UAG: RF1 only activity) (Fig 5B). These variants were cleanly inserted at the endogenous *sigB* operon in wild-type and in an RF-inducible strain (RF1/PrmC and RF2, with frameshift removed, under orthogonal promoters, Materials and Methods). Under various RF expression conditions (endogenous, RF2 knockdown, RF1/PrmC knockdown), we monitored mRNA levels of two responsive $\sigma^B$-dependent genes by RT–qPCR (*ywzA* and *ygxB*, highlighted in Fig 3A, Materials and Methods). Basal $\sigma^B$ activity at endogenous RF expression differed slightly for the different *rsbV* stop variants, likely as a result of perturbed ORF positioning. Strikingly, we observed strong $\sigma^B$ activation only in strains with the *rsbV* stop cognate to the RF perturbation (Fig 5C, raw data in Dataset EV8). Consistently with previous experiments, RF2 knockdown led to an increase (> 8×) in the expression of our $\sigma^B$ reporter genes in the *rsbV*$_{UGA}$ variant (Fig 5C, top row). In addition, induction of reporter genes was observed for the *rsbV*$_{UAG}$ variant only under RF1/PrmC knockdown (> 50×, Fig 5C, bottom row). No induction in either RF perturbation was observed for the *rsbV*$_{UAA}$ variant (Fig 5C, middle row), in accordance with the UAA stop codon being RF agnostic. The *rsbV*$_{UAA}$ variant was confirmed functional (> 500× induction, indistinguishable from the wild-type *rsbV*$_{UGA}$ variant following 5 min of 4% v/v ethanol stress, Dataset EV8). Hence, our stop codon swapping experiment demonstrates that systems-wide susceptibility to a distal molecular perturbation can be encoded through a single-sensitive *cis*-element and that such susceptibility can be entirely rewired with minimal genetic changes.

## Discussion

We have shown that quantitative genome-wide measurements in conjunction with precision fitness quantification can be used to reconstruct chains of events across biological scales relating a molecular perturbation (change in expression of a specific protein) to a whole-cell phenotype (growth rate).

We confirm with high precision that in *B. subtilis,* release factors are expressed at levels that maximize cell growth rate (Fig 2), a proposition that is often suggested but rarely tested. Our data thus corroborate several previous lines of evidence suggesting that RF expression might be precisely tuned. First, it was found that the relative expression between RF1 and RF2 correlates with stop codon usage between different species (Korkmaz *et al,* 2014; Wei *et al,* 2016). For instance, *B. subtilis* has a higher abundance of RF1 and more frequent UAG usage compared with *E. coli*, suggesting that RF1's expression set point meets translational demand (Materials

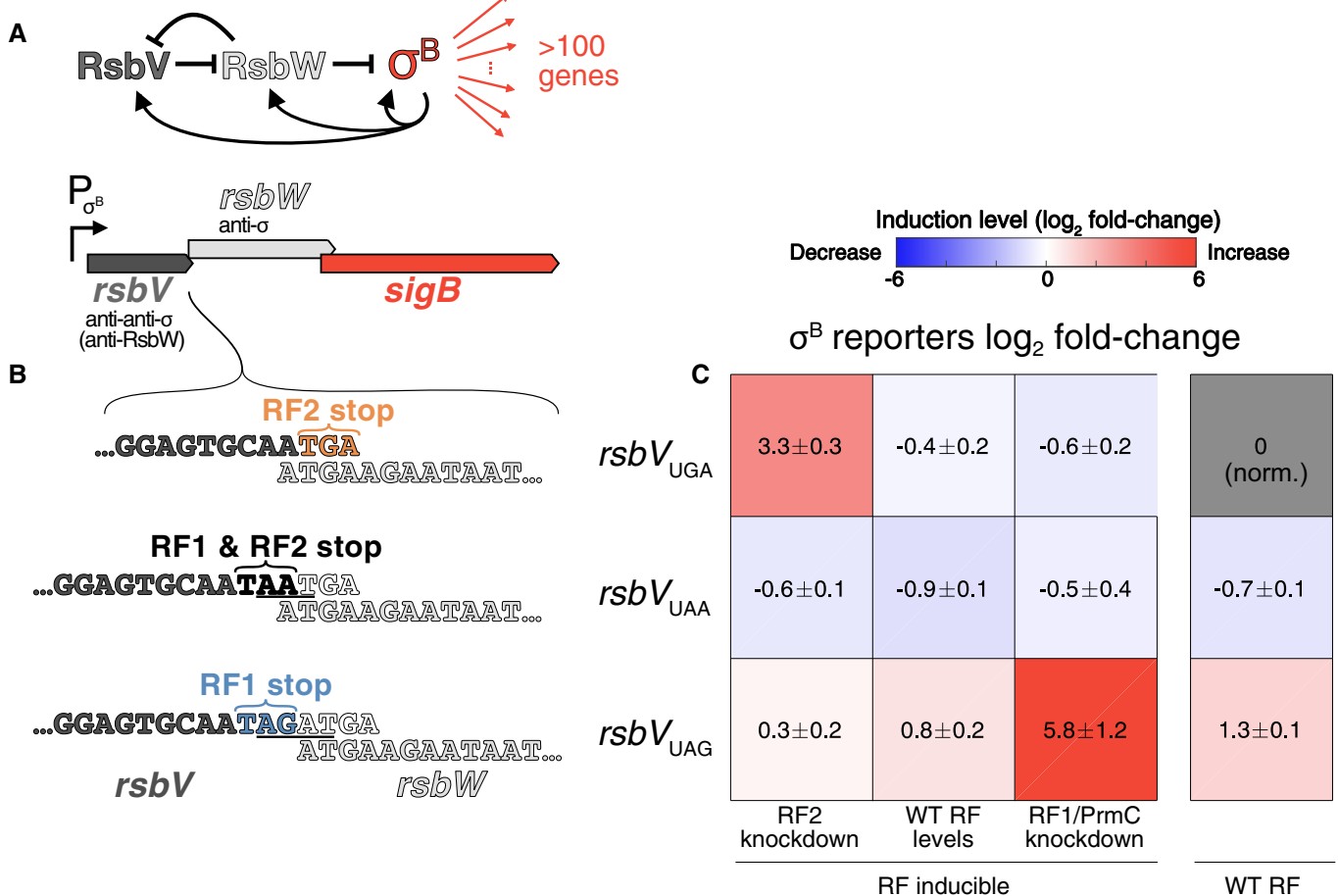

**Figure 5. Swapping a single stop codon rewires $\sigma^B$ induction susceptibility to RF perturbation.**

A Schematic of simplified $\sigma^B$ regulatory architecture and operon structure.

B Different *rsbV* stop codon variants considered. Top row shows endogenous configuration *rsbV*$_{UGA}$, with RF2-dependent stop codon and A<u>UGA</u> open reading frames overlap. Middle row shows the UAA RF-agnostic variant *rsbV*$_{UAA}$, obtained by adding AAT (underlined). Bottom row shows the RF1-dependent UAG allele *rsbV*$_{UAG}$, obtained by inserting AGAT (underlined).

C Average of reporter gene $\log_2$ fold-change compared with wild-type for $\sigma^B$ reporter genes (*ywzA* and *ygxB*, highlighted in Fig 3A) as quantified by RT–qPCR (2–3 independent biological replicates for each allele/condition, ± indicates s.e.m. over replicates for the two reporter genes, raw data in Dataset EV8, Materials and Methods) showing strong induction of reporter genes in the RF perturbations cognate to the stop codon of the *rsbV* variant. Rows correspond to *rsbV* stop variant in panel (B) and columns to RF expression conditions. RF-inducible measurements were done in strains with orthogonally tunable RF1/PrmC (IPTG) and RF2 (xylose; Materials and Methods).

Data information: See also Appendix Fig S5.

and Methods). Second, the gene encoding RF2 has a broadly conserved UGA-based frameshift event that autoregulates the expression based on its own activity (Craigen *et al*, 1985; Craigen & Caskey, 1986; Baranov *et al*, 2002). Interestingly, there are no reports of RF1 autoregulation to our knowledge, and we found that ectopic over- or under-expression does not affect its own promoter activity (Appendix Fig S4). Therefore, a lack of autoregulation does not necessarily indicate that cells are less sensitive to small perturbations on its expression.

Upon perturbation, we find that important portions of the selective pressures on peptide release factor abundances are through idiosyncratic activation of distal endogenous regulatory programs (Fig 6). By analogy with interactions between residues constraining the evolution of individual proteins (Bridgham *et al*, 2009; Pollock

*et al*, 2012; Shah *et al*, 2015; Starr *et al*, 2018), we term this strong dependence of enzymes' expression-fitness landscapes on the network of genetic interactions "regulatory entrenchment" (Fig 6).

Despite our focus on translation termination factors, which we anticipated to have limited impacts on gene expression (translation termination being non-limiting on mRNAs), pleiotropic effects related to activation of specific regulons still dominated the connection between RF expression and fitness. As an example of molecular determinants shaping expression-fitness landscapes, we found that a single *cis*-element (identity of the *rsbV* stop codon, Fig 5B and C) mechanistically led to RF-specific expression imbalance (altered stoichiometry of $\sigma^B$ regulators RsbW and RsbV, Fig 4D). This imbalance was amplified to global transcriptome remodeling through the activation of $\sigma^B$ (Figs 3A and EV3C). These changes ultimately had

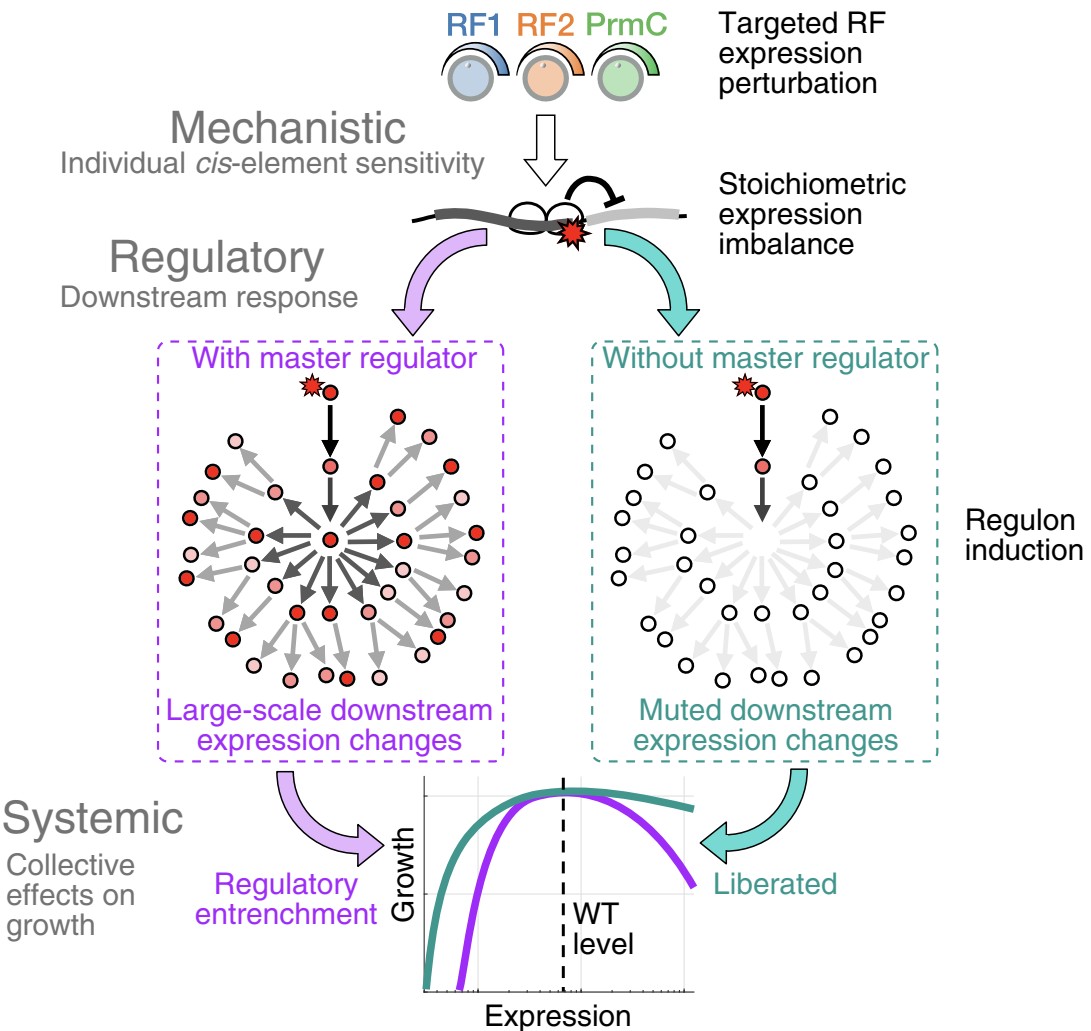

**Figure 6.  Regulatory entrenchment in gene expression-fitness landscapes.**

Perturbation of specific enzyme expression can lead to changes in the expression of a subset of genes through the sensitivity of *cis*-regulatory elements. These expression imbalances can further reverberate in the network of regulatory interactions in the cell through focal points such as master regulators, which control the expression of multiple genes (purple box). In the absence of these regulators, downstream responses are muted (teal box). Pleiotropic expression changes exacerbate the fitness defect of perturbing the original enzyme. The cell's susceptibility to expression is then not simple relation of the decreased enzymatic flux, but also depends on the aggregated impacts regulatory susceptibilities of the whole cell. Cells can be liberated from regulatory entrenchment by removing nodes in the network.
Data information: See also Figs EV3 and EV4.

systemic impacts (growth defects by proteome compression, Figs 3E and EV3G and O). Although the exact cause of $\sigma^B$ activation due to PrmC overexpression remains to be determined, deletion of the *sigB* gene was sufficient to flatten the landscape in two directions of the RF expression space (PrmC overexpression Figs 3G and EV3H and RF2 knockdown Fig EV3Q).

The adaptive nature of $\sigma^B$ activation upon RF stress remains a possibility, but the lack of corresponding fitness advantage in our experiments suggests otherwise (Figs 3G and EV3H). Bioinformatic analysis of homologous $\sigma^B$ operons shows some conservation in the *rsbV/rsbW* ORF overlap (Appendix Fig S5, Materials and Methods). Overlap of these genes could however be driven by the requirement of a precisely tuned expression stoichiometry (Igoshin *et al*, 2007; Locke *et al*, 2011), enabled by translational coupling via the −4 nt

start/stop overlap configuration A**UGA** and additional yet-unknown features that renders them particularly sensitive to RF2 perturbations. In such scenario, susceptibility to RF perturbation would emerge as an evolutionary byproduct of selective pressures that operate on other features of the system (Gould & Lewontin, 1978).

As an additional example of regulatory entrenchment in our data, the sharp decrease in fitness upon RF1 knockdown (Fig 2E) was associated with gene expression changes characteristic of the motile-to-sessile bistable switch in *B. subtilis* (Kearns & Losick, 2005; Chai *et al*, 2009) (Fig EV4). Given the complexity of the regulation of motility genes (Mukherjee & Kearns, 2014), we have not been able to identify a unique plausible molecular link between expression changes and RF1 knockdown (Appendix Supplementary Methods). Further, the relationship between the transition to sessile

state and the plummeting cell growth rate also remains elusive and might not be completely causative. Indeed, we found that the most prominent transcriptome changes following RF1 knockdown do not fully explain the growth defect, as a strain deleted for *sigD*, which mimics the motility and biofilm regulon expression in RF1 knockdown (Kearns & Losick, 2005), only contribute to a < 10% decrease in growth rate (doubling time 21 ± 1 min wild-type, 23 ± 1 min Δ*sigD*). These underscore the challenges of reconstructing causal features underlying expression-fitness landscapes, even with detailed under-the-hood information about the cell's internal state.

Our case study highlights that exhaustive information at multiple levels, down to the response of all *cis*-regulatory elements, might be required to achieve fully predictive models given how easily perturbations propagate across scales in biological systems. A lack of global characterization of the susceptibilities and interaction of components in regulatory networks currently constitute an important bottleneck in systems biology, although at-scale methods are being developed to dissect genetic networks (preprint: Calderon *et al,* 2020; Muller *et al,* 2020). Various strategies can now generate expression-fitness landscapes for a large number of genes in parallel, for example using suites of promoters (Keren *et al,* 2016), genome-scale library of inducible gene expression (preprint: Arita *et al,* 2021), or tunable CRISPR perturbations (Hawkins *et al,* 2020; Jost *et al,* 2020; Mathis *et al,* 2021). Together with the advent of single-cell transcriptomics in bacteria (Blattman *et al,* 2020; Imdahl *et al,* 2020; Kuchina *et al,* 2020), these methods open the possibility of dissecting the molecular underpinnings of expression-fitness landscapes genome-wide, and to comprehensively identify instances of regulatory entrenchment.

We anticipate that master regulators will be common focal points of regulatory entrenchment, as seen here with general stress factor $\sigma^B$ and motility factor $\sigma^D$, due to the responsiveness of their activation and their direct impact on a large number of genes. As such, successful predictions will presumably be more easily achieved in near-minimal organisms with limited regulation (Hutchison *et al,* 2016; Baby *et al,* 2018; Matteau *et al,* 2020). Given the broadly conserved expression stoichiometry of enzymes across evolution (Lalanne *et al,* 2018), we speculate that endogenous expression levels satisfy generic optimization principles, such as flux maximization under resource allocation constraints (Kurland & Ehrenberg, 1987). However, since diverse bacteria harbor distinct *cis*-elements, operonic organization, and regulatory architectures, the shape of expression-fitness landscapes away from the optima might generally differ as a result of regulatory entrenchment.

# Materials and Methods

## Strains

RF-inducible and control strains used for the competition experiments are listed in Dataset EV1. Strains used for expression profiling (RNA-seq) were GLB115 (wild-type), GLB426 (independently inducible RF2 and PrmC), GLB430 (independently inducible RF2 and PrmC, *sigB* knockout), GLB434 (control with blank ectopic constructs at *amyE* and *lacA*), GLB438 (independently inducible RF1 and PrmC), GLB442 (independently inducible RF1 and PrmC, *sigB* knockout), and GLB446 (control with blank ectopic constructs

at *amyE* and *levB*). Of note, our laboratory strain of *B. subtilis* subsp. 168 has the *swrAA*⁺ allele (Calvio *et al,* 2005) (as determined by RNA-seq reads mapped to *swrAA*) leading to expression of motility gene in exponential growth, in contrast to other subsp. 168 and laboratory strains (Kearns *et al,* 2004).

Strains with *rsbV* UGA (wild-type), UAA, and UGA variants with endogenous release factors' loci are GLB115, GLB 450, and GLB451, respectively. Strains with *rsbV* stop UGA, UAA, and UAG variants with independently inducible RF2 (xylose promoter) and RF1/PrmC/YwkF (the native RF1 operon under IPTG inducible promoter) are, respectively, GLB452, GLB453, and GLB454. See Dataset EV1.

Strains for the CRISPRi experiment were a kind gift of Prof. Jason M. Peters (CAG74829: sgRNA to *prfA*, CAG74815: sgRNA to *prfB*) (Peters *et al,* 2016). These have sgRNA under constitutive expression (P$_{veg}$ promoter) inserted at *amyE* (with chloramphenicol resistance cassette), and catalytically inactive Cas9 under xylose promoter integrated at *lacA* (with erythromycin resistance cassette).

Strain with matched ribosome profiling and Rend-seq data for translation efficiency calibration of RF1 and RF2 expression constructs was GLB452 (see details below). Other datasets with matched ribosome profiling and Rend-seq data used solely for calibration between the transcriptome and proteome synthesis fractions for $\sigma^B$ and translation sectors involved strains GLB372 (deletion of *ylbF*) and GLB455 (inducible green and red fluorescence proteins).

The *sigD* null strain used for comparison of growth to wild-type was DS323 from (Kearns & Losick, 2005), provided by the Grossman laboratory.

A list of plasmids and oligos used for molecular cloning can be found in Dataset EV2.

## Strain construction

Standard protocol relying on natural competence of *B. subtilis* and on recombination via flanking homology sequences were used (Harwood & Cutting, 1990). Molecular cloning of plasmids and recombinant DNA relied on isothermal assembly.

Two types of genetic modification were made: ectopic integration via flanking homology with resistance cassettes (Guérout-Fleury *et al,* 1995) (for inducible expression systems), and markerless genetic modification (clean genetic modification with no resistance marker, used for barcode diversity generation, endogenous RF copy deletions, and *rsbV* stop codon variant generation). The pminiMAD2 strategy (single cross-over integration of modification plasmid followed by counterselection based on a temperature-sensitive origin of replication) developed by Patrick and Kearns (Patrick & Kearns, 2008) was used for markerless cloning in *B. subtilis.*

Cloning in RF-inducible strains was performed with inducers appropriate for endogenous expression (25 μM IPTG, 0.035% w/v xylose) throughout growth and plating steps. In all instances of markerless cloning, full swap out of the cloning cassette was confirmed by PCR with primers outside the homology regions. All strains were confirmed by Sanger sequencing of appropriate PCR products obtained from genomic DNA of candidate clones.

### RF-inducible strains

As our inducible expression systems (ectopic insertion cassettes illustrated in Fig EV1B and C), we used promoters P$_{xyl}$ (with

repressor XylR, derived from pDR160 (Bose & Grossman, 2011), a kind gift from Prof. Alan Grossman) responsive to xylose, and $P_{spankHy}$ (with repressor LacI, derived from pDR111 (Quisel *et al,* 2001), a kind gift from Prof. Rich Losick) responsive to IPTG. The $P_{xyl}$ was used without the *xylA* mini-ORF for the expressed proteins. The $P_{xyl}$ promoter has a unimodal response upon xylose induction (Peters *et al,* 2016). The response of LacI to IPTG is also unimodal in the absence of the lac permease LacY (Marbach & Bettenbrock, 2012) which is *de facto* not present in *B. subtilis*.

We extensively tested stability of induction of both $P_{xyl}$ and $P_{spankHy}$ promoters at different cell densities by RT–qPCR (data not shown). These confirmed that expression from these ectopic constructs was constant throughout and beyond the range ($OD_{600}$ between 0.05 and 0.4) of cell densities attained during our competition experiments and expression measurements. Projections of attained RF1, RF2, and PrmC in the 3D subspace of expression (Fig EV1E) are shown in Fig EV1F (quantified by RNA-seq calibrated with ribosome profiling) for our inducible strains across conditions profiled, showing ability to orthogonally modulate RF levels.

To generate RF-inducible strains, ectopic expression constructs with the appropriate inducible gene copies were first inserted, following by the markerless deletion (pminiMAD2 strategy) of the endogenous copies. The strain with independently inducible RF1 and PrmC was challenging to generate because of high toxicity of *B. subtilis*' RF1 (alone) overexpression plasmid in *E. coli* (not shown). Hence, to generate an expression construct with RF1 individually tunable, we integrated the full RF1 operon under a $P_{spankHy}$ promoter and then deleted *prmC* and *ywkF* from both the endogenous locus and the ectopic operon. For RF2-inducible constructs, the autoregulatory frameshift in RF2 (Craigen *et al,* 1985; Craigen & Caskey, 1986) was removed by removing a single nucleotide (nucleotide T at position 73 in the *prfB* gene) in the ectopic construct. Once the RF-related genetic modifications were completed, strains were barcoded at the *amyE* locus for competition experiments (details below).

RF1 (gene *prfA*) and RF2 (gene *prfB*) are in operons and co-transcribed with other genes. Gene *ywkF* (unknown function) is downstream and co-transcribed with *prfA* and *prmC* (gene encoding PrmC). Upon deletion of endogenous *prfA* and *prmC*, the promoter endogenous transcript was retained to preserve *ywkF* expression. Upon removal of *prmC* only, no measurable effect on the levels of the mRNAs of *prfA* and *ywkF* mRNAs were observed (e.g., RNA-seq data from strain GLB426). RF2 (gene *prfB*) is co-transcribed downstream of gene *secA*. A partial transcription terminator separates the two genes (as determined from Rend-seq (Lalanne *et al,* 2018)). Upon deletion of endogenous *prfB*, no effect on the mRNA level of upstream gene *secA* was detectable (e.g., RNA-seq data from strain GLB426). The gene downstream of *prfB*, *yvjA* (unknown function), which normally not co-transcribed with *prfB* (beyond low abundance transcription termination read-through mRNAs) in endogenous conditions, had increased expression in the strains with deletion of the *prfB* gene, because of an increase in the *prfB* transcription terminator readthrough (despite its sequence being preserved by our *prfB* deletion). Identifying the causes of this change in readthrough the *prfB* transcription terminator following upstream gene deletion is beyond the scope of the current work.

## Barcode diversity generation

We used an 8 bp region in the *amyE* (positions 328,634–32,8641 on chromosome, annotation NC_000964.3, which was part of one of the homology regions used for ectopic cassette integration) as our location to introduce chromosomal barcodes. These barcodes served as a way to mark different genotypes (or redundant strains with the same genotype) in our competition experiments.

To generate different chromosomal barcodes at those positions, we created a library of "barcode swapping" plasmids. These were pminiMAD2 (Patrick & Kearns, 2008) variants with homology to *amyE* with a set of random nucleotides at the barcode positions. We generated two such plasmid libraries: one for modification in the endogenous *amyE* genotype and one for modification in a genotype where the *amyE* region had been disrupted by an ectopic expression construct (e.g., Fig EV1B and C). Importantly, the barcode was internal to the *amyE* region, such that readout of the region was possible using identical primers for all barcoded strains (see Fig EV2B for schematics of the steps involved barcode readout). The resulting genetic changes for a successful barcode swap were the modification of only the 8 bp inside *amyE*. Following transformation of the plasmid libraries and counterselection, clones with modified barcodes were identified by Sanger sequencing of appropriate PCR products. This strategy was employed to barcode all strains used in our competition experiments.

## Growth conditions

All growth was performed on a tabletop orbital shaker with vigorous aeration (220 rpm) at 37°C.

### Fitness landscape measurement

Growth for concurrent fitness and expression (RNA-seq) quantification (Figs 2, 3, EV3 and EV4) was carried out in MC complete (MCC) defined medium (Parker *et al,* 2020) with glycerol as the main carbon source (1% w/v). Doubling time of wild-type *B. subtilis* in this medium (as determined by manual exponential growth curve) was 21 ± 1 min. Inducers xylose and IPTG were added to various concentrations to modulate RFs expression levels in our inducible strains (details in Dataset EV3). MCC medium was conditioned to decrease the lag phase upon back dilution of cultures during competition experiments. To condition the medium, seed cultures of wild-type *B. subtilis* (GLB115) were started from freshly streaked LB plates in unconditioned MCC medium. Once pre-culture reached $OD_{600} \approx 0.2$, the cultures were diluted to $OD_{600} = 0.001$ in the MCC medium to be conditioned. At $OD_{600} = 0.15$, medium was vacuum filtered through a coarse filter (450 nm nitrocellulose). Given prior evidence of weak pausing on asparagine/proline codons (data not shown) from ribosome profiling in conditioned medium, asparagine and proline were re-supplemented by an additional 1× their initial concentration and the resulting medium was filter-sterilized.

To minimize sources of variability, competition experiments and growth in monoculture for expression quantification were performed on the same day, with the same batch of medium, on the same shaker, from the same pre-cultures. On the morning of the day prior to experiment, the ≈ 20 strains involved in each experiment were streaked out (see Dataset EV3 for a list of strains included in

each competition experiment). Later in the day, seed cultures were initiated from these fresh plates (individual growth tube for each barcoded strain) in conditioned MCC medium with inducer concentration appropriate for approximate wild-type expression of the release factors (25 μM IPTG, 0.035% w/v xylose). The next morning (day of the experiment), for competition experiments, strain pre-cultures were pooled from the overnight cultures at 1:1 ratios. The pool was diluted to $OD_{600} \approx 0.01$ in conditioned MCC medium with 25 μM IPTG and 0.035% w/v xylose until recovery to $OD_{600} = 0.1$. At that point, the pool was diluted to $OD_{600} = 0.004$ in 16 ml (125 ml flasks) of medium across multiple flasks corresponding to the different inducer concentrations in the experiment. Once $OD_{600}$ reached 0.1 ($\approx$ 7.8 generations) in each flask, each pool was diluted into another flask (16 ml of medium) of the same medium (pre-warmed, with same inducer concentration, and shaking) to $OD_{600} = 0.004$. At the time of first transfer, 8 ml of culture was harvested (in 0.8 ml of 10:1 ethanol/phenol stop solution). This first harvest constituted time 0 of the competition experiments. Subsequently, each time $OD_{600}$ reached 0.1, cultures were diluted back to $OD_{600} = 0.004$ and 8 ml were harvested as above. This procedure was maintained for 4 transfers (5 harvest points including time 0), or for about 30 generations. Care was taken to maintain cultures in exponential growth throughout by these frequent dilutions at low optical density ($OD_{600} \leq 0.1$).

Monocultures were started in parallel to the start of the competition for expression profiling by RNA-seq. Chosen strains (one of the barcoded strains from each specified genotypes, see Dataset EV3) were diluted from the overnight culture (same culture as that used for pooling of competition experiments) to $OD_{600} = 0.01$ in conditioned MCC medium (with 25 μM IPTG and 0.035% w/v xylose) for recovery and grown until $OD_{600}$ reached 0.1. At that point, the cultures were diluted to $OD_{600} = 10^{-4}$ in 25 ml of medium (125-ml flasks) in individual flasks with various inducer concentrations. 10 ml of cells were harvested (in 1 ml of 10:1 ethanol/phenol stop solution) once $OD_{600} = 0.1$.

In total, 3 days of competition experiments were performed, with nine inducer conditions to move along the RF1 dimension (experiment E2, see Dataset EV3, Fig 2A), nine inducers conditions to move along RF2 dimensions (experiment E1, see Dataset EV3, Fig 2 B), and 7 + 7 inducer conditions to move along the PrmC dimension (with wild-type RF2 level, and overexpression of RF2; experiment E3, see Dataset EV3, Fig 2C and D).

### CRISPRi RF depletion

CRISPRi experiments (Figs 4 and EV5) were carried out in LB with xylose as the inducer of the dCas9 construct (0.04% and 0.05% xylose w/v for RF1 and RF2 strains, respectively). Overnight cultures (initiated from freshly streaked plates) in LB without xylose were back diluted to $OD_{600} = 3 \times 10^{-4}$ in LB with xylose and were grown until $OD_{600} = 0.3$.

### RT–qPCR experiments

For *rsbV* stop codon switch experiments (Fig 5), seed cultures were started from freshly streaked plates in conditioned MCC medium with inducers (25 μM IPTG, 0.035% w/v xylose). Once $OD_{600}$ reached 0.1, cultures were back diluted to $OD_{600} = 10^{-4}$ in conditioned MCC medium with various inducer concentrations.

Cells were harvested at $OD_{600} = 0.1$. Measurement of $\sigma^B$ activation level under lower RF2 expression (no xylose, and strain with two copies of the xylose repressor gene) were performed as above in conditioned MCC medium with 1% glycerol (no xylose, 25 μM IPTG).

The experiment to test the *rsbV* UAA variant function by ethanol stress was performed in LB. Cultures were started from freshly streaked plates and back diluted to $OD_{600} = 3 \times 10^{-4}$. Cells (wild-type GLB115, and *rsbV* UAA variant GLB450) were harvested once at $OD_{600} = 0.25$ (no stress). Ethanol was then added to 4% v/v, and cells harvested 5 min later (with ethanol stress).

### Matched Rend-seq/ribosome profiling datasets

Matched Rend-seq and ribosome profiling datasets used for translation efficiency calibration and conversion between transcriptome and proteome fraction were collected by diluting overnight cultures to $OD_{600} = 3 \times 10^{-4}$ in the medium respective growth medium (see Dataset EV4 for details of strains and growth media), and harvested at $OD_{600}$ between 0.15 and 0.3.

### Gene expression measurements

Expression datasets generated and used in this work can be found in Dataset EV4, together with the corresponding gene-by-gene expression quantification. Oligonucleotides used for library preparation and qPCR primers are listed in Dataset EV2.

### RNA-seq

As our primary method to quantify gene expression genome-wide, we used a previously reported RNA-seq strategy (Parker *et al*, 2019). Briefly, 10 ml of culture at $OD_{600} = 0.1$ were collected and mixed with 1 ml of 10:1 ethanol/phenol stop solution by rapid inversion. Cells were pelleted at 3,000 rcf for 10 min at 4°C. The supernatant was decanted and cell pellets stored at −80°C until RNA extraction. RNA was extracted using the RNeasy plus kit with gDNA eliminator column (Qiagen) following manufacturer's instructions. The concentration of RNA in samples was assayed by Qubit (Thermo Fisher) from 10× dilution (RNA BR). Samples were diluted to concentration of 660 ng/μl. rRNA was removed using the MICROBExpress kit (Thermo Fisher), following manufacturer's instruction but loading 2.5 μg of RNA to the reaction, using 1/8× reaction volumes, and 1 μl of capture oligo mix. Following isopropanol precipitation, samples were resuspended in 11 μl 10 mM Tris 7, and the concentration determined by Qubit (Thermo Fisher, RNA BR). For each sample, 250 ng of rRNA removed RNA was diluted in 40 μl of 10 mM Tris 7. The RNA was incubated at 95°C for 2 min and brought to ice. 4.4 μl of 10× RNA fragmentation reagents (Thermo Fisher) were added, and the mix placed at 95°C for 1 min 45 s, following which 5 μl of 10× stop solution was added. The fragmented RNA was purified with Oligo Clean and Concentrator columns (Zymo) and eluted in 17 μl 10 mM Tris 7. 3 μl of T4 PNK master mix (2 μl 10× PNK buffer, 0.5 μl SUPERase·In [Thermo Fisher], 0.5 μl T4 PNK enzyme [NEB]) was added per sample, mixed, and incubated for 60 min at 37°C, followed by 10 min at 75°C. 10 μl of PolyA master mix (3 μl 500 mM KCl, 3 μl 10 mM ATP, 2 μl 5× FS buffer (from SuperScript III enzyme), 0.5 μl SUPERase·In, 1 μl water, and 0.5 μl *E. coli* poly A polymerase [NEB]) was then added per sample, mixed, and incubated at 37°C for 30 min,

followed by 10 min at 75°C. 1 µl of 25 µM indexed poly-dT reverse transcription primers (each sample having its own index, allowing pooling of samples) was added per sample, mixed, and incubated at 65°C for 5 min. The samples were returned to ice, and 9 µl of reverse transcription master mix was added per sample (3 µl 0.1 M DTT, 2 µl 10 mM dNTP mix, 2 µl 5 × FS buffer [from SuperScript III enzyme], 1 µl water, 0.5 µl SUPERase·In, and 0.5 µl SuperScript III [Thermo Fisher]), mixed, and incubated at 50°C for 60 min, followed by 75°C for 10 min. Samples with different indices from reverse transcription primers were then pooled and the RNA hydrolyzed by adding 0.1× volume of 1 M NaOH, followed by 15-min incubation at 95°C. cDNA in the range 100–120 nt was then size selected on a 10% TBU polyacrylamide gel (Thermo Fisher), gel extracted, and isopropanol precipitated. The cDNA was then resuspended in 20 µl of 10 mM Tris 8. 10 µl of the size selected cDNA was mixed with 5 µl 100 µM ligation adapter oDP214, 3 µl water, 5 µl 10× T4 DNA ligase buffer, 5 µl 5 M betaine, 20 µl poly(ethylene glycol) 8000, and 2 µl T4 DNA ligase (NEB), mixed thoroughly, incubated at 16°C for 10 h. The reaction was cleaned up (Oligo Clean and Concentrator, Zymo) and eluted in 10 µl 10 mM Tris 8. The ligated cDNA was size selected (135–155 nt) on a 10% TBU polyacrylamide gel, gel extracted, and precipitated. Low cycle number PCR was performed with Q5 DNA polymerase (NEB; standard reaction mix) with primers oDP007 and oDP010 (98°C for 30 s denaturation, with cycles of 10 s at 98°C, 10 s at 60°C, and 7 s at 72°C). 5–7 cycles of amplification were usually sufficient. The final library was size selected on 8% TBE polyacrylamide gel, gel extracted, and isopropanol precipitated. Details of the final amplicon for the RNA-seq libraries can be found in Dataset EV4.

Sequencing data were processed as follows. Poly A tails were stripped (retaining read portion upstream of 18 A residues, for reads without 18 A residues, reads ending with at least 16 A residues were also retained), and resulting reads (> 14 nt in length) aligned to the *B. subtilis* genome using bowtie (option v1 k1) (Langmead *et al*, 2009). The 3′ ends of mapped reads were summed at each genomic position. To quantify gene expression, the average read density (excluding 20 bp gaps from the annotated start and end of genes, i.e., from start +20 to end −20) was computed and is reported in rpkm (reads per kilobase per million mapped reads) by normalizing by the total read counts not mapping to rRNA or tRNAs. Quantification for each gene in each dataset can be found in Dataset EV4.

mRNA levels from RF-perturbed cells were compared with average mRNA levels from unperturbed datasets (average across biological replicates). Following the lack of observed changes in strains with blank expression cassettes controls (e.g., Fig EV2I and J), these datasets were also included in estimating the average unperturbed mRNA levels for each gene (see Dataset EV4 for list).

The current RNA-seq cDNA library preparation method is highly reproducible. For example (Appendix Fig S1A), pairwise comparisons across our six wild-type datasets (with different inducer concentrations which should only affect the expression of two genes, *xylA* and *xylB*, samples E1_C1_GLB115, E1_C5_GLB115, E1_C9_GLB115, E2_C1_GLB115, E2_C5_GLB115, E2_C9_GLB115, see Dataset EV3 for details) showed a $R^2$ for log-transformed mRNA level comparison across genes (> 100 reads mapped) of 0.99. Across pairs, the median 10$^{th}$ to 90$^{th}$ percentile in fold-change in mRNA levels (denoted FC$_{10}^{90}$) fell between 0.86 and 1.16 (genes with > 100 reads mapped). The RNA-seq method used for our

expression-fitness landscape mapping also compares favorably with a different library preparation approach (Rend-seq (Lalanne *et al*, 2018)) with different 3′ and 5′ adapter molecular cloning approaches: 10$^{th}$ and 90$^{th}$ percentile of fold-changes between the two methods for genes with > 100 reads mapped ranged between 0.78 and 1.23 from libraries prepared starting with the same RNA material (data not shown). The above RNA-seq method is also highly reproducible, with biological replicates harvested and with libraries prepared on different days from different biological samples (same strain and growth conditions) having 10$^{th}$ and 90$^{th}$ percentile fold-changes of 0.87 and 1.13 for genes with > 100 mapped reads (data not shown).

### Ribosome profiling

Ribosome profiling and downstream quantification were performed as described in (Li *et al*, 2014; Lalanne *et al*, 2018) with slight modifications. Given the absence of a substantial 5′ ramp in *B. subtilis* and the limited impact of correction from pausing Shine–Dalgarno-like sequences internal to genes (Lalanne *et al*, 2018), we directly used the ribosome footprint density to as the per-gene protein synthesis rates (excluding the first and last 20 bp from the gene for quantification). In line with recent observations (Mohammad *et al*, 2019), we included a wide range of footprint sizes (14–44 nt). Pile-up files were generated from center-mapped footprints.

Quantification of protein synthesis by ribosome profiling is highly reproducible (Li *et al*, 2014). For example (Appendix Fig S1B), from data from this work, biological replicates (samples from wild-type ribo_MCC1_GLB115/ribo_MCC2_GLB115, and samples from wild-type with inducible fluorescent proteins ribo1_GLB455/ribo2_GLB455, see Dataset EV4) show an $R^2$ of log-transformed ribosome footprint density across all genes with > 100 footprint reads mapped of 0.98 and 0.99, respectively, and a corresponding fold-change 10$^{th}$ to 90$^{th}$ percentile between 0.83–1.17 and 0.91–1.12, respectively. We do caution that the density of ribosomes at or after stop codons can be sensitive to several experimental conditions, such as harvest speed and footprinting parameters.

Stop codon usage was estimated as the synthesis fraction of all genes with the specified stop codon (or stop tetranucleotide, i.e., including the 4$^{th}$ nucleotide after the stop), which accounts for the flux through the stop codon (as opposed to relative a gene count). The stop codon usage in *B. subtilis* is 0.888 for UAA, 0.064 for UAG, and 0.049 for UGA, compared with, respectively, 0.888, 0.015, and 0.097 in *E. coli*. Concomitant to increased usage of UAG, the abundance of RF1, cognate to stop UAG, is higher in *B. subtilis* (estimated proteome fraction 0.085) than in *E. coli* (estimated proteome fraction 0.01).

Translational readthrough (Fig 4B) (Mangano *et al*, 2020) was estimated as the ratio of footprint density downstream of the stop codon (+5 to +45 nt) to that inside the body of the gene. Only isolated genes (closest upstream and downstream co-directional genes > 55 bp away) with a density of > 0.1 footprint/nt were considered. We note that some genes have estimated readthrough > 1 due to, e.g., unannotated ORFs, low read counts, or repetitive regions, which can inflate the readthrough score in some rare occasions.

### RT–qPCR experiments

10 ml of culture were collected and mixed with 1 ml of 10:1 ethanol/phenol stop solution by rapid inversion. Cells were pelleted

                                                          

at 3,000 rcf for 10 min at 4°C. The supernatant was decanted and cell pellets stored at −80°C until RNA extraction. RNA was extracted using the RNeasy plus kit with gDNA eliminator column (Qiagen) following manufacturer's instructions, and eluted in 10 mM Tris 7. The concentration of RNA in samples was assayed by spectrophometry (ND-1000, Thermo Fisher) from a 10× dilution. Samples were diluted to concentration of 1 µg/µl. 1 µl of RNA was added to 1 µl of 100 µM random hexamers, heated for 5 min at 65°C, and placed on ice. 8 µl of RT master mix (1 µl 10× reaction buffer, 0.5 µl 10 mM dNTPs mix, 0.5 µl M-MuLV reverse transcriptase (New England Biolabs), and 6 µl DEPC-treated water) was added and mixed by pipetting. The mix was incubated 5 min at 25°C, 60 min at 42°C, and 20 min at 65°C. For each RNA sample, reactions with and without reverse transcriptase were run in parallel. The RNA was subsequently hydrolyzed by adding 2 µl of 1 M NaOH and heating to 95°C for 5 min. The solution was neutralized with 2 µl of 1 M hydrochloric acid, and the reaction volume was brought to 100 µl with DEPC-treated water. For qPCR, 5 µl of Kappa SYBR green master mix, 2 µl of 1 µM forward and reverse primers, and 3 µl of diluted cDNA (above) were mixed. Reactions were monitored on a LightCycler 480 system (Roche), and $C_t$ values obtained from the maximal second derivative method from the machine software. For each sample, multiple qPCR primer pairs were run in parallel. Samples were run on technical triplicates on 384 well plates, and large outliers among triplicates (difference in $C_t$ value > 0.2 from technical replicates, typically < 5% of wells) were excluded. Large differences in $C_t$ values (> 7) for reactions with and without reverse transcriptase were confirmed for each sample/primer pairs in each experiment. The mean $C_t$ value among technical replicates was calculated and used for expression quantification. Primers to constitutively expressed genes *gyrA* were used for loading normalization. For a gene of interest, the mRNA levels relative to that of *gyrA* were calculated as $2^{Ct_{gyrA}-Ct_{oi}}$ (qPCR primer efficiencies were estimated to not differ substantially from 2 by serial dilution experiments, data not shown). Quantification from RT–qPCR experiments can be found in Dataset EV8.

### Rend-seq/Ribosome profiling calibration

Ribosome profiling was performed as detailed previously (Li *et al*, 2014; Lalanne *et al*, 2018). Rend-seq (end-enriched RNA-seq) was performed as described (Lalanne *et al*, 2018). Given the small effect of ribosome pausing corrections and lack of 5′–3′ ramp in *B. subtilis* (Lalanne *et al*, 2018), relative protein synthesis rates were estimated directly as the mean ribosome footprint read density over genes (excluding 20 bp gaps from the annotated start and end of genes, i.e., from start +20 to end −20). mRNA levels from Rend-seq were quantified as the read density over gene bodies (excluding 20 bp gaps from start/end of genes as above). Rend-seq and ribosome profiling expression quantification can be found in Dataset EV4.

To calibrate the expression of RF1 and RF2 in our RF-inducible strains, we used concurrent ribosome profiling and RNA-seq data (using Rend-seq library preparation strategy) in strain GLB452 (see Dataset EV1). GLB452 has RF2 without frameshift under $P_{xyl}$ and the full RF1 operon (RF1, PrmC, and YwkF) under $P_{spankHy}$. Endogenous copies of *prfA* (RF1 gene) and *prfB* (RF2 gene) are deleted. The translation efficiency (per mRNA rate of translation, denoted TE) for the exogenous copies of RF2 and RF1 was determined as ribosome profiling rpkm divided by Rend-seq rpkm, and used to derive a fold-change TE compared with wild-type mRNAs for these genes, leading to $3.0 \pm 0.4$ and $0.66 \pm 0.05$ for RF2 and RF1, respectively (error bar from standard error of the mean from 5 different RNA-seq/ribosome profiling datasets pairs at different induction levels for the constructs). Expression (measured by RNA-seq) for RF1 and RF2 in comparison plots was corrected by the above fold-change in TE above.

Slight differences between strains used for ribosome profiling (above) and the RF-inducible strains used for competition experiments are as follows. RF1 expression construct in strain GLB438 and GLB442 (used for competition experiment and fitness landscape determination) is different in that genes *prmC* and *ywkF* are not present (in-frame deletion for *prmC*) from the ectopic transcript under $P_{spankHy}$. The 5′ UTR and ribosome-binding site for the RF1 gene are identical to those of GBL452 (in which translation efficiency calibration was performed, detailed above). We therefore use the GBL452 calibrated TE for the *prfA* mRNA in strains GLB438 and GLB442. Strains with inducible RF2 expression in competition (strains GLB426, GLB430) use the exact same construct as GLB452, with expression of RF2 coming from the same exogenous mRNA. The translation efficiency of the *prfB* mRNA calibration from GLB452 is thus directly applicable to GLB426 and GLB430. For exogenous inducible PrmC copies (under $P_{spankHy}$ in GLB426 and GLB430, and under $P_{xyl}$ for GLB438 and GLB442), we do not have corresponding TE estimates from ribosome profiling. The endogenous translation efficiency of PrmC is 0.51 (38[th] percentile). As a parsimonious estimate, we assume no change in the translation efficiency of our exogenous construct, which is consistent with the optimum in the expression-fitness landscape corresponding to close to the expected endogenous position (Fig 2G).

To convert our mRNA level quantification from RNA-seq to a calibrated proteome fraction (as shown on the fitness landscape plots, e.g., Fig 2E–H,), we take the wild-type proteome fraction (estimated as the proteome synthesis fraction, see below), multiplied by the fold-change in mRNA level (directly measured by RNA-seq), and finally multiplied by the fold-change in translation efficiency (as discussed above) for the exogenous constructs.

### Estimating the proteome mass fraction

The conversion from ribosome profiling data to proteome mass fraction relies on ribosome profiling providing an accurate estimate of protein synthesis and on proteins being generally stable over the duration of a cell doubling time. In bacterial cells, the overwhelming majority of proteins have degradation rates small compared with the dilution arising from cell growth (Larrabee *et al*, 1980). For ribosome footprint density to provide an accurate measurement of protein synthesis, two assumptions need to be met: (i) the majority of ribosomes that initiate translation complete the full peptide and (ii) the average translation elongation rate is uniform across transcripts. Prior comparison to measured abundances in the literature, and assessment of protein production among stoichiometric obligatory complexes, have shown ribosome profiling to provide an accurate and precise measure of protein synthesis (Li *et al*, 2014). Here, the proteome synthesis fraction for a given gene was calculated from ribosome profiling data as the estimated synthesis rate multiplied by the gene size (proportional to the total number of ribosome footprint reads mapping to the gene), divided by the sum of this quantity over all genes. For

a fully stable proteome, the protein synthesis fraction then equals the proteome mass fraction.

### Regulon transcriptome fraction

The list of annotated regulons with gene members in *B. subtilis* was downloaded from SubtiWiki (Zhu & Stülke, 2018). The transcriptome fraction for each gene was determined as the total number of RNA-seq reads mapping to member genes and divided by the total number of reads not mapping to rRNA or tRNAs. Regulon transcriptome fraction was equal to the sum of transcriptome fraction for genes annotated as members of the regulon. The list of highlighted genes in specific regulons (Figs 3A and B, 4D and E, EV2I and J, EV3A–D, J, O and P, and EV4C–E) can be found in Dataset EV5.

### Estimating regulon proteome fraction

We compiled acquired datasets for which Rend-seq (mRNA level quantification) and ribosome profiling (measure of protein synthesis) were obtained from the same culture. These were used to, respectively, quantify the transcriptome fraction $\psi$ and proteome fraction $\phi$ (see "Estimating the proteome mass fraction" above for the conversion from ribosome profiling data to proteome fraction) for regulons of interest, here the $\sigma^B$ regulon and the set of mRNA translation proteins.

For translation proteins, the relationship between proteome and transcriptome fraction was well captured by $\phi_R = \alpha \psi_R$ (with $\alpha \approx 1.1$) across our matched Rend-seq/ribosome profiling datasets, although the range of observed values for $\phi_R$ was limited (from 0.30 to 0.42). This conversion factor was used to estimate the proteome fraction from transcriptome fraction (excluding RF1, RF2, and PrmC to avoid confounding the contribution of these factors resulting from overexpression) determined by RNA-seq.

For the $\sigma^B$ regulon, the relationship capturing the matched datasets was $\Delta\phi_{SigB} = \alpha\Delta\psi_{SigB}$ ($\alpha \approx 1.41$), where $\Delta\phi_{SigB} := \phi_{SigB} - \phi^\circ_{SigB}$ and $\Delta\psi_{SigB} := \psi_{SigB} - \psi^\circ_{SigB}$ are the excess proteome and transcriptome fractions from the respective basal values $\phi^\circ_{SigB}$ and $\psi^\circ_{SigB}$. Hence, we find empirically that $\phi^\circ_{SigB} \approx \psi^\circ_{SigB}$, but $\Delta\phi_{SigB} > \Delta\psi_{SigB}$. This indicates that upon induction of $\sigma^B$ regulon genes, the average translation efficiency of the mRNAs of regulon genes increases. This coarse-grained observation is mechanistically corroborated, as we found numerous examples for which the production of new mRNA isoforms as a result of increased in activity in alternative promoters driven by $\sigma^B$ leads to large increase in translation efficiency compared with the basal isoform expressed by housekeeping factor $\sigma^A$ (preprint: McCormick et al, 2021). To estimate the proteome fraction to the $\sigma^B$ regulon from multiplexed RNA-seq transcriptome fraction, we use $\phi_{SigB} = \alpha\Delta\psi_{SigB} + \phi^\circ_{SigB}$, with $\alpha \approx 1.41$.

### Growth rate measurement by pooled competition

We use high-throughput sequencing to quantify the relative proportion of different barcoded strains in our competition experiments, relying on the relative proportion of different barcode reads in amplicon pools as a proxy for the relative proportion of cells harboring each barcode. To readout the barcodes, we use two steps of PCR (Fig EV2B), appending an index encoded in primers at each step, allowing us to pool and multiplex measurement from different samples. The final amplicon is compatible with RNA-seq and ribosome profiling libraries, allowing pooling on the same sequencing lane. The approach is similar to Bar-seq (Smith et al, 2009) and derivatives, but instead of sequencing a highly complex pool at two time points, we consider a pool of limited number of barcodes ($\approx 20$ strains) at multiple time points in order to improve precision of the readout (Parker et al, 2020). Compared with previous high-precision approaches to measuring fitness based on luminescence (Kishony & Leibler, 2003; Kavčič et al, 2020) or flow cytometry (Gallet et al, 2012; Duveau et al, 2017, 2018), our method could in principle be further improved by increasing the number of barcoded strains per genotype, which provide semi-biological replicates within each pooled experiment.

### Chromosomal barcode readout

8 ml of culture (from the pool of barcoded strains in competition) was collected at $OD_{600} = 0.1$ and mixed with 0.8 ml of 10:1 ethanol/phenol stop solution by rapid inversion. Cells were pelleted at 3,000 rcf at 4°C for 10 min and the supernatant decanted. The cell pellets were stored at −80°C until DNA extraction. To extract genomic DNA (gDNA), we used the Promega Wizard gDNA extraction kit, following the manufacturer's protocol (scaling reagents volumes by 1/3×). The resulting gDNA was quantified on Qubit (Thermo Fisher, dsDNA BR). Samples were diluted to a concentration of 100 ng/μl. The first PCR was performed using the standard Phusion DNA polymerase (NEB) reaction to final volume of 12.5 μl, with 775 ng of template gDNA and an indexed primer with 16 nt UMI (10 variants of PCR1_UMI_FOR with different indices) and common reverse primer (oJBL124). These primers have annealing regions outside the barcodes in *amyE* common to all strains, see Fig EV1B and C. The parameters for the first PCR (PCR1, Fig EV2B) were as follows: initial denaturation at 98°C for 30 s, followed by three cycles of denaturation (98°C for 10 s), annealing (63°C for 30 s), and elongation (72°C for 10 s). Following the first PCR, the reaction mixes were put on ice. Reactions with different indices from this PCR were pooled, cleaned up (Zymo Clean and Concentrator 5), and eluted in 100 μl 10 mM Tris 8. To get rid of residual primers and gDNA, we further applied each cleaned-up pool from the first PCR to select-a-size columns (Zymo), adding 120 μl ethanol to the eluate of the first column. Final elution was made in 36 μl 10 mM Tris 8. The second PCR (PCR2, Fig EV2B) was carried out with primers annealing to common regions of amplicons from the first PCR and served to append both a second index and adapters required for sequencing on Illumina platform (Truseq primers). The PCR followed standard Phusion DNA polymerase (NEB) reaction in 12.5 μl. The parameters for the second PCR were as follows: initial denaturation at 98°C for 30 s followed by a variable number of cycles (6–7 cycles usually sufficient) of denaturation (98°C for 10 s), annealing (60°C for 30 s), and elongation (72°C for 6 s). The final amplicon libraries (211 bp) were size selected by gel extraction on 8% TBE polyacrilamide gels (Thermo Fisher).

The structure of the final amplicon, with primers, can be found in Dataset EV6. Indices for the PCRs for our competition experiments can be found in Dataset EV6.

### Estimation of relative fitness

Assuming no interaction between cells in a pool, and exponential growth throughout the experiment, the number of cells for strain

*mut* will grow according to $N_{mut}(t) = N_{mut}^0 2^{t/\tau_{mut}}$, where $\tau_{mut}$ is the doubling time of strain *mut* in this particular environment. The ratio of number of cells for strain *mut* to wild-type (WT) then obeys:

$$\log_2\left(\frac{N_{mut}(t)}{N_{WT}(t)}\right) = \log_2\left(\frac{N_{mut}^0}{N_{WT}^0}\right) + \frac{t}{\tau_{WT}}\left(\frac{\tau_{WT}}{\tau_{mut}} - 1\right).$$

We take the number of generations to be $T_{gen} := t/\tau_{WT}$, and define the relative fitness coefficient $s := \frac{\tau_{WT}}{\tau_{mut}} - 1 = \frac{\lambda_{mut}}{\lambda_{WT}} - 1$ (where $\lambda := \log(2)\tau^{-1}$ is the growth rate). We assume the number of reads corresponding to a strain barcode (after collapsing reads with the same UMI's from the primers of first PCR) $R_{mut}$ to be proportional to the number of cells corresponding to that strain in the pool (by a factor that is constant throughout the experiment), i.e., $R_{mut}(t) \propto N_{mut}(t)$, then:

$$\log_2\left(\frac{R_{mut}(t)}{R_{WT}(t)}\right) = \log_2\left(\frac{R_{mut}^0}{R_{WT}^0}\right) + sT_{gen}.$$

Hence, we take relative fitness (between a strain pair) as $1 + s$, where $s$ is the slope of the $\log_2$ ratio of the barcode counts as a function of number of generations. Given that most of the strains in our pool have close to wild-type growth rates, we estimate the number of generations as $-\log_2$ of the dilution factor between harvest points (e.g., $-\log_2$ (70 μl/16 ml) $\approx 7.8$ generations for our standard dilution protocol.

UMI collapsed barcode counts (for each strain, condition, and time point) are listed in Dataset EV6, together with estimated $s$ for each pair of strains for each condition profiled (from the linear fit described above).

The error on the slope (as estimated by least-square) sets the precision of our fitness measurement. Assuming independent identically distributed $\log_2$ ratio measurements with standard deviation $\sigma_{\log_2 r}$, we can derive from the expression of the slope under least-square regression that the standard deviation the slope $s$, $\sigma_s^{LS}$, is (under even time sampling) for total number of generations $T_{gen}^{tot}$ and $n_t$ sampling points:

$$\sigma_s^{LS} = \sqrt{12\frac{n_t-1}{n_t+1}}\frac{\sigma_{\log_2 r}}{\sqrt{n_t}T_{gen}^{tot}} \approx \sqrt{12}\frac{\sigma_{\log_2 r}}{\sqrt{n_t}T_{gen}^{tot}},$$

where the approximation is for large $n_t$. Preliminary tests with mixing of strains at fixed ratios and technical replicate (gDNA extraction and amplicon preparation) had shown $\sigma_{\log_2 r} \approx 0.2$. Based on these, we chose to perform experiments for $\approx 30$ generations, with $n_t = 5$ samplings to have a precision of better than 1% ($\sigma_s^{LS} \approx 0.6\%$). This estimate turned out to be close to our empirical precision based on comparison of isogenic pairs (Fig EV2E).

Figure EV2C and D shows representative examples of the fitness estimation procedure for pairs of strains in a given condition (wild-type to wild-type in Fig EV2C, and wild-type to RF-inducible in Fig EV2D). Importantly, redundantly barcoded strains with the same genotype provide semi-biological replicates in the same competition pool, improving our precision. For example, four of the 21 strain pairs are shown for redundantly barcoded WT to WT comparisons in Fig EV2C. We report the interquartile range ($25^{th}$–$75^{th}$ percentile) of the isogenic pairs $s$ in our main figures (e.g., error bars in Fig 2E–H), which are typically smaller than the plotted symbol. The measured values for all pairs of strains for the comparison of fitness

upon RF2 knockdown with and without *sigB* are shown in Fig EV3Q.

For comparison of the relative growth rates for RF-inducible strains with and without gene *sigB*, the small fitness effect of removing the *sigB* gene in an otherwise wild-type background was subtracted. Specifically, we report $s_{RF inducible, \Delta sigB} - \langle s_{\Delta sigB}\rangle$, with $\langle s_{\Delta sigB}\rangle$ the average fitness defect of the *sigB* deletion across all conditions in a given competition experiment (there were slight variations in $\langle s_{\Delta sigB}\rangle$ from one competition experiment to another).

## Quality control of growth measurement

Numerous experiments were performed to assess precision and accuracy of the barcode readout protocol and fitness measurement, as well as biological impact, independent of RF perturbations, of our ectopic expression cassettes and resistance markers.

### Barcode readout cross-talk assessment

The two-step PCR protocol to generate our amplicon library allows us to append two sets of indices on our samples, permitting multiplexing amplicon sequencing of various conditions. Such pooling of samples prior to PCR2 can introduce cross-talk between indices and barcodes. To characterize such cross-talk, we spiked-in genomic DNA extracted from individual-barcoded strains not present in the sequenced pool at specific combinations of PCR indices (Datasets EV6 and EV7). Bleed through of these barcodes to different PCR indices provides a measure of cross-talk arising (for example) from carryover-indexed primers from the first PCR to the second PCR.

Datasets EV7 shows the results for barcode cross-talk for competition experiments. Rows correspond to indices from the first PCR (PCR1) and columns to indices from the second PCR (PCR2). PCR1-indexed samples are pooled prior to PCR2. The tables show the number reads mapping to the barcode for the spiked-in genomic DNA (each table corresponds to a different spike-in) with indices from PCR1 and PCR2. Highlighted positions in the table correspond to positions where corresponding spike-in was added. Read counts at non-highlighted positions correspond to cross-talk

Two types of cross-talk can be distinguished: (i) intra-PCR2 pool (appearance of spiked-in barcode at different PCR1 indices inside pools where the spike-in was added) corresponding to reads not highlighted in a column with a highlighted cell and (ii) inter-PCR2 pool (appearance of spiked-in barcode at PCR1 indices in pools where the spike-in was not added) corresponding to reads in a column with no highlighted cell. Interpool cross-talk was very low, with relative number of reads incorrectly mapping per spike-in barcode being $< 5 \times 10^{-6}$. Intrapool cross-talk was higher, but $> 99\%$ (and typically higher) of spike-in barcode reads mapped to the correct pair of PCR indices. Prior experiments without the additional primer clean-up before the select-a-size column purification after PCR1 suggested higher intrapool cross-talk, reaching $> 10\%$ in some cases (data not shown), possibly from carryover primers from PCR1. The additional clean-up step strongly reduced barcode cross-talk.

### Accuracy of barcode readout

Mixing of two strains with different at predetermined ratios followed by our barcode readout procedure recovered the expected ratios across four orders of magnitudes (Fig EV2H).

### Accuracy of fitness measurement

The accuracy of the method was determined by comparing the measured $s$ from competition experiment with the growth rate measured via a "manual" growth curve (measurement of optical density versus time) from the monoculture from which cells were harvested for multiplexed RNA-seq. Dataset EV7 lists the comparisons. Conditions for which $s$ from competition experiment were $|s| > 0.2$ (large growth defect) and with three or more $OD_{600}$ data points within the range 0.005 to 0.12 (reliability of manual doubling time estimate) were retained for comparison. The inferred $s$ from the growth curve was calculated as $s := \frac{\tau_{WT}}{\tau_{mut}} - 1$, with $\tau_{WT} = 21 \pm 1$ min and $\tau_{WT}$ from the linear fit of log-transformed $OD_{600}$ data points. Range for the doubling time measurements is estimated from bootstrap subsampled slopes. Overall, most measurements agreed within error, with the manual measurements being much less precise, and with agreement being better for larger growth defect (as expected given the noise in the manually estimated $s$ for small growth defect).

### Precision of fitness measurement

A stringent measure of precision of our relative growth measurement from competition is the distribution of measured relative fitness $s$ arising from isogenic strains (apart from having a different chromosomal barcode) across our conditions. Given that most genotypes are redundantly barcoded at least 4 times (except $\Delta sigB$, with two barcodes), this provides us with 6 pairwise comparisons per condition per genotype pairs. Under the assumption that the identity of the 8 bp barcode does not affect fitness, $s$ should be 0 for these strain pairs. We emphasize that in addition to noise in the readout, non-zero $s$ could come from accumulated deleterious/beneficial mutations in the course of the barcoding cloning procedure and pre-cultures prior to competition experiments. Hence, the distribution of $s$ for isogenic pairs (examples of such isogenic pair competitions are shown in Fig EV2C) corresponds to a lower bound on the precision of the measurement readout itself. Across $n = 1,253$ such pairwise comparisons spanning all inducer conditions and experiments, we find $\sigma_s = 0.6\%$ (median $|s| = 0.3\%$), with the full distribution of isogenic $s$ (Fig EV2E), shown as inset in Fig 2H. We take the resolution of our fitness measurement to be $\pm 2\sigma_s = 1.2\%$ (gray shading in Fig EV2E).

### Impact of ectopic expression cassettes

Our RF expression constructs involve the disruption of endogenous loci (*amyE*, *lacA*, and *levB*), together with the addition of exogenous genes (various resistance cassettes and repressors) which could contribute to fitness defect either through cost of expression or the specific activity of these genes (although none of the resistance cassettes used in our inducible RF strains modify endogenous cell machinery). We constructed and redundantly barcoded control strains with "blank" ectopic expression cassettes, which were identical to the cassettes used to drive inducible expression of RFs, but without any genes under the inducible promoters. These control strains were included in all our competition experiments. Two suites of such control strains were made: (i) with blank insertions at *amyE* and *lacA* (GLB434–437 series), and (ii) with blank insertions at *amyE* and *levB* (GLB446–449 series). See strain details in Dataset EV1.

Both transcriptome characterization (representative comparisons with wild-type in Fig EV2I for GLB434, and Fig EV2J for GLB446 showing little genome-wide changes) and fitness measurements (Fig EV2F and G shows the distribution of $s$ for these strains compared

with wild-type across our experimental conditions) showed little impact of these ectopic cassettes. In particular, we did not see trend with $s$ as a function of increasing inducer concentration. Overall, our control strains showed growth defects of about 1% or smaller, close to the resolution of our measurement, suggesting little fitness defects arising from the disruptions to the genome and expression of exogenous proteins (resistance and repressor proteins). Given these small fitness defects for control strains, we compared our RF-inducible strains directly with wild-type because of the higher number of barcoded wild-type strains, improving our precision.

### Assessing the impact of the lag phase

Our competition experiments relied on periodic dilutions of cultures to maintain approximate steady-state exponential growth, as opposed to using an automated continuous culture device such as a turbidostat (Toprak *et al*, 2013; Wong *et al*, 2018; Schober *et al*, 2019). While conditioning our growth medium reduced lags between dilutions, and dilutions were made at the low $OD_{600}$ of 0.1, a short lag was still observed at each dilution. To confirm that the observed growth defects originated from the exponential growth rate differences as opposed to slight differences in lag time between strains, we performed two series of competition experiments with the same number of total generations, but different number of dilutions (few large dilutions vs. many small dilutions). The two different dilution scenarios consisted of the same total number of generations of growth, but additional lag periods for the scenario with many small dilutions. Specifically, the first scenario consisted of four dilutions of 1,000× (40 generations total) with harvesting at each dilution and at $t = 0$ (scenario large dilutions $L$), and the second scenario consisted in eight dilutions of 31.6× (40 generations total) with harvesting at $t = 0$ and every other two dilutions (scenario small dilutions $S$). These experiments were performed with a single barcoded strain compared with wild-type (one barcode) and were with a different RF-inducible system (GLB452, same as *rsbV* stop codon switching and ribosome profiling calibration), where RF2 was under $P_{xyl}$ and the full RF1 operon (RF1, PrmC, and YwkF) was under $P_{spankHy}$. 9 different conditions with these two dilution protocols were considered, spanning different regions of the (RF1, RF2, PrmC) expression space. Results for measured $s$ for the two different dilution scenarios, compiled in Dataset EV7, showed near-complete agreement between the two dilution scenarios, supporting that possible slight differences in recovery from the lag phase upon RF perturbation did not contribute to the measured growth defects, even for large perturbations.

## Details on the $\sigma^B$ regulon

### Selection of reporter genes in $\sigma^B$ regulon

To select reporter genes to assess $\sigma^B$ regulon activity by reverse transcription quantitative PCR (RT–qPCR), we identified a list of genes annotated as members of the $\sigma^B$ regulon which were consistently most highly induced transcriptionally in our RNA-seq datasets. *gsiB* was the most strongly induced $\sigma^B$ gene, but its sequence contained multiple duplicated regions, making its quantification by RT–qPCR unreliable. *ywzA* and *ygxB* were two highly and consistently induced genes in the regulon (highlighted dark red in Fig 3A). These genes were selected as targets to monitor induction of the regulon for the *rsbV* stop codon switching experiment.

### Bioinformatic analysis of $\sigma^B$ operons

To search for other *sigB* operons in diverse species, we scanned the representative and reference genomes from the RefSeq database (Tatusova *et al*, 2016) with protein blast (evalue cutoff $10^{-7}$) to the three amino acid sequences of RsbV, RsbW, and $\sigma^B$ from *B. subtilis*. We retained species with hits to all three proteins, extracted positions of homologous proteins on chromosomes, and identified connected clusters of hits based on a permissive spatial cutoff of 100 bp (distance between start to end of genes). Retaining connected spatial clusters with the three proteins, and also filtering based on conserved gene order and orientation, led to 95 final candidate $\sigma^B$ operons (the operon with $\sigma^B$ homolog with highest identity to *B. subtilis'* $\sigma^B$ was retained in species with multiple candidate operons). Candidates are listed in Dataset EV9. Final candidates are summarized in Appendix Fig S5. For these *sigB* operon candidates: $18/95 = 19\%$ had the <u>AUGA</u> overlap between *rsbV* and *rsbW*. In comparison, only 7% of co-directional gene pairs (226/3,042) have that arrangement in *B. subtilis* (Fig EV5J). Further, $42/95 = 44\%$ of *sigB* operon candidates had any form of coding sequence overlap between *rsbV* and *rsbW*, compared with 17% (521/3,042) across co-directional gene pairs in *B. subtilis*.

### CRISPRi depletion of RF1/PrmC and RF2

As an orthogonal system to perturb expression of RF, we performed experiments in CRISPRi (dCas9 targeted to a specific locus by sgRNA, blocking transcription and decreasing gene expression by up to 100×). Strains targeting *prfB* (strain CAG74815) and *prfA* (strain CAG74829) (Peters *et al*, 2016), a kind gift of Prof. Jason M. Peters, were used. Given the operon structure of RF1 and PrmC (co-transcribed), the knockdown of RF1 also leads to knockdown in PrmC.

Characterization of the induction property of the xylose promoter (driving dCas9 expression in the CRISPRi strains) in LB using RT-–qPCR revealed around 20× induction at 0.04% w/v xylose as the $OD_{600}$ of the culture increased from 0.1 to 0.3 (data not shown). The physiological causes of such induction at intermediate $OD_{600}$ are beyond the scope of the current work, but could be due to consumption of residual glucose present in LB, alleviating competitive inhibition of the xylose repressor (Dahl *et al*, 1995). This transient induction was used to generate an acute, but non-steady-state, knockdown of RF1/PrmC and RF2. Following dilution to $OD_{600} = 3 \times 10^{-4}$ in LB with xylose (0.04% for RF1, 0.05% for RF2), and cells were harvested for Rend-seq and ribosome profiling (flash filtration) at $OD_{600} = 0.3$.

Quantification of mRNA levels and protein synthesis in these conditions revealed over 60× knockdown for both RF1 and RF2 (RF2 mRNA level quantified by average read density after the position targeted by the sgRNA: positions 3,627,150 to 3,628,026 in NC_000964.3), Fig 4C and D. Together with these RF-specific perturbations, our protein synthesis measurements showed large-scale induction of $\sigma^B$ upon RF2 knockdown (Fig 4D), and mild *eps* gene induction in RF1/PrmC knockdown (median fold-change increase of 2.9 compared with wild-type, not highlighted in Fig 4C), consistent with remodeling in the gene expression program observed in our steady-state measurements with our RF-inducible strains.

While RF abundances are challenging to quantify from synthesis rates in non-steady-state conditions, queuing was observed at RF-specific stop codons by ribosome profiling (Figs 4A and EV5A–C),

suggesting high depletion in RF. Metagene stop codon traces (Fig 4 A) were obtained as follows: For each stop codon (UAA, UAG, UGA), all genes expressed to at least 0.5 ribosome footprint read/nt were retained and their mapped footprint reads (center-mapped (Li *et al*, 2014)) traces (normalized by the mean density in the first half of the gene) were aligned by the stop codon. The median at each position over all such aligned traces constituted metagene plot (Fig 4A). Each stop codon shows elevated ribosome density in wild-type. However, for RF1/PrmC and RF2 knockdown, elevated densities following periodic patterns of $\approx 25$ nt (ribosome footprint size) appeared upstream of stop codons specific to the knocked-down RF, Fig 4A (bottom two rows), likely corresponding to ribosome queues forming as a result of slow translation termination. All ribosome footprint traces for genes meeting the read density threshold were also visualized as a heat map, Fig EV5A–C. More prominent queues were observed for genes with higher translation efficiency TE (as determined from wild-type data from Lalanne *et al*, (2018), TE being proportional to the rate of ribosome initiating on individual mRNAs), consistent with a simple stochastic theory of queue formation (schematic Fig EV5H) (Bergmann & Lodish, 1979; Mitarai *et al*, 2008), Appendix Supplementary Methods.

The ribosome profiling data under acute RF knockdown was used to assess expression stoichiometry of co-directional gene pairs (Fig 4 E). To do so, the ribosome footprint density ratio for selected gene pairs was calculated in wild-type and under RF knockdown. Gene pairs, restricted to pairs with > 10 reads mapping to both genes in the two conditions, were then stratified based on stop codon identity and distance between gene pairs. The distribution of fold-change of the expression stoichiometry (down/up ribosome footprint density) under RF perturbation vs. WT was then recorded under different stratifications. Figure 4E shows the fold-change distribution for co-directional gene separated by < 30 bp, and stratified by the stop codon of the upstream gene. It shows a modest but clear decrease in the expression of gene downstream of stop codons cognate to the RF perturbation. Additional control analyses were performed to assess the validity of this effect. First, gene pairs within 30 bp but stratified by the stop codon of the downstream gene did not show this effect, Fig EV5D. In addition, gene pairs separated by more than 30 bp stratified by the upstream stop codon did not show this effect, Fig EV5E. Performing the same analysis (stratified by upstream stop codon, within 30 bp) but from Rend-seq data (same experiment) also did not show this effect (Fig EV5F), suggesting that the decrease in expression came from changes in translation initiation, and not mRNA levels. Analyses considering fold-changes in gene expression (compared with wild-type, normalized by the median fold-changes of all genes with sufficient coverage, > 10 reads mapped in both conditions, irrespective of stop codons) stratified by stop codon (Fig 4F for ribosome profiling, Fig EV5G for mRNA levels) did not show substantial stop codon-specific effects.

### Statistical analysis

To assess significance of the stop codon-specific trends observed in our expression datasets for acute RF depletion by CRISPRi (Figs 4E and F, and EV5D–G), we generated random reshufflings of stop codon categories (i.e., the identities of the gene's stop codon were randomly permuted). For each stop codon reshuffling, we calculated the median of fold-change for the (reshuffled) RF-specific stop

cognate to the RF perturbation and the median of the fold-change for the (reshuffled) RF-agnostic stop UAA. The ratio of these median fold-changes constituted the effect size. The *P*-values were taken as the fraction of stop codon reshuffling for which the effect size was more pronounced (smaller ratio of fold-changes) than the non-reshuffled list.

To assess the significance of the fitness rescue upon *sigB* deletion for the maximum steady-state knockdown of RF2 (Fig EV3Q), we randomly sampled (with replacement) 12 (out of total 96) pairwise fitness differences at the non-maximal RF2 knockdown. The median of these sampling was compared with the median of the sampled (12 samplings with replacement of the 12 pairwise fitness values) fitness differences under maximal RF2 knockdown. The *P*-value was taken as the fraction of random samplings for which the median of the fitness difference for non-maximal RF2 knockdown exceeded the median of the fitness difference of maximal.

## Data availability

The datasets and computer code produced in this study are available in the following databases:
- RNA-seq, Rend-seq, and ribosome profiling data, in addition to sequencing data from amplicons in competition experiments: Gene Expression Omnibus (GSE162169; https://www.ncbi.nlm.nih.gov/geo/query/acc.cgi?acc = GSE162169)
- Data analysis computer scripts: Code EV1 included as a supplementary file in the current work.

**Expanded View** for this article is available online.

## Acknowledgements

We thank J. M. Peters for providing CRISPRi strains, A. Grossman for providing the ΔsigD strain, members of the G.-W.L. and A. Grossman laboratories for critical discussions, and T. Starr for introducing us to the concept of entrenchment. We thank L. Herzel for critical comments on the manuscript. This research was supported by NIH grant R35GM124732, the NSF CAREER Award, the Smith Odyssey Award, the Pew Biomedical Scholars Program, a Sloan Research Fellowship, the Searle Scholars Program, the Smith Family Award for Excellence in Biomedical Research, NSERC doctoral Fellowship, and HHMI International Student Research Fellowship to J.-B.L., NIH T32GM007287 to D.P.

## Author contributions

J-BL and G-WL designed experiments and analysis; J-BL performed experiments, collected data and performed analysis; J-BL and DJP designed and optimized competition experiments; DJP established defined growth medium for *B. subtilis* and RNA-seq method; J-BL and G-WL wrote the manuscript with comments from DJP.

## Conflict of interest

The authors declare that they have no conflict of interest.

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
