## [Review Process File · Molecular Systems Biology]

Spurious regulatory connections dictate the expression-fitness landscape of translation factors

Jean-Benoit Lalanne, Darren Parker, and Gene-Wei Li
DOI: [10.15252/msb.202110302](https://doi.org/10.15252/msb.202110302)

Corresponding author(s): Gene-Wei Li (gwli@mit.edu)

Review Timeline:	Transfer from Review Commons:	19th Feb 21
	Editorial Decision:	23rd Feb 21
	Revision Received:	12th Mar 21
	Accepted:	16th Mar 21

Editor: Maria Polychronidou

Transaction Report:

This manuscript was transferred to EMBO Molecular Systems Biology following peer review at Review Commons.

Review #1

1. How much time do you estimate the authors will need to complete the suggested revisions:

Estimated time to Complete Revisions (Required)

(Decision Recommendation)

Less than 1 month

2. Evidence, reproducibility and clarity:

Evidence, reproducibility and clarity (Required)

The manuscript of Lalanne and coworkers address the cellular responses to varied translation termination factor expression in *Bacillus subtilis*. The authors set-up a system to fine-tune the expression of release factor RF1, RF2 as well as PrmC that post-translationally modifies RF1/RF2 to maximize their catalytic hydrolysis activity. They then monitor the fitness costs associated with overexpression or depletion of the factor by following the changes in growth rate. The set-up is nicely illustrated in Figure 1. The results in Figure 2 show that overexpression of RF1 and RF2 has relatively modest effect on the growth rate compared to overexpression of PrmC that leads to dramatic growth rate reduction.

By contrast, depletion of RF1 has a strong negative influence on fitness, whereas a similar level of depletion of RF2 had little influence on fitness.

PrmC overexpression appears to be correlated with the induction of the sigmaB regulon, however, the authors do not manage to ascertain why this is. By contrast, RF2 depletion also results in the induction of the sigmaB regulon and the authors demonstrate convincingly that this is due to a termination defect within the *rsbQ-rsbV* operon that contains an overlapping start-stop AUGA

A few points that the authors might consider discussing

1. The natural abundance of each RF in bacteria in relation to the usage of different stop codons in different organisms.
2. The role of the frameshifting mechanism in RF2 and how then RF1 levels are regulated.
3. The authors observe queuing in front of the relevant stop codons upon RF depletion, however, do not discuss about readthrough events, which are usually competing with termination. Surprisingly, in this context the authors don't discuss the work from Mankin and coworkers showing sequestration of RFs from termination by peptides such as apideacin leads to translational readthrough.
4. The efficiency of translation termination is well-known to be dependent on the context of the stop codon. Do the authors also observe such a trend. Especially, UGAC for RF2, one would expect to observe high levels of readthrough upon RF2 depletion.

3. Significance:

Significance (Required)

Overall, the experiments are clearly performed and beautifully illustrated. Clearly, a lot of work has gone into this study but the end message that the cell regulates carefully RF concentrations is not surprising. Especially given that RF2 carefully regulates its own levels using an autoregulatory frameshifting mechanism. The major finding that the *rsbQ-rsbV* operon with the RF2 dependence leading to induction of the *sigmaB* regulon is in the end rather trivial since these regulators depend on RF2 for termination. Therefore, this manuscript is unlikely to have general interest to people in the translation field (such as myself) but rather those working in the field of synthetic biology.

Review #2

1. How much time do you estimate the authors will need to complete the suggested revisions:

Estimated time to Complete Revisions (Required)

(Decision Recommendation)

Less than 1 month

2. Evidence, reproducibility and clarity:

Evidence, reproducibility and clarity (Required)

In this paper, the authors use a combination of RNA sequencing, ribosome profiling and measurements of cellular composition and growth rate to gain insight into the multi-scale effects that perturbations to translation termination factors have on general physiological states and reproductive fitness using *Bacillus subtilis* as their model organism. Specifically, they find that perturbing the expression levels of peptide chain release factors in any direction has a negative effect on growth-rate. This negative effect was not due to a direct impact of the gene on the cell, but instead due to a chain of regulatory interactions that cause the activation of the general stress regulon. This leads to upregulation of a large chunk of the genome and an indirect impact on the expression of all other genes. Critically, the knock-on effects observed for the specific perturbations studied suggest that it may be difficult to predict expression-fitness landscapes of a cell, without carrying out a detailed mapping of all genes and the cell's physiological state.

Overall, the core findings in the paper are well justified by the data presented and the experiments appear to have been rigorously carried out. My only concern is that it is unclear if biological replicates of the ribosome profiling were performed. Also,

biological replicates are mentioned for the RNA-seq data, but no data is shown. Even a simple graph demonstrating the expression levels across these would be useful to be assured of no issues in reproducibility given the complex processing of the data involved. Related to this, I see no mention of data availability in the paper. For this study to be useful to others, providing the raw data (unprocessed) would be essential (ideally in a public repository).

The presentation of the work is excellent, with very clear figures and text that helped guide the reader through the results. There were a few minor comments:

1. Abstract: "in bacterium *Bacillus subtilis*" should read "in the bacterium *Bacillus subtilis*".
2. Page 4: "found that under numerous ways" should read "found that under the numerous ways".
3. The authors mention that changes in the expression level of RF1 impacted motility and biofilm genes, but not how this impacts fitness. Would they be able to experimentally identify origin of RF1 growth defects in the same way they did for PrmC? This is not essential for the main findings but would help strengthen the work.
4. It is difficult to know how generalisable the findings of this work are due to the very limited scope. It could be helpful for the authors in the discussion to consider and comment on how such approaches might be scaled-up to enable broader and more general studies of expression-fitness landscapes and where they will find most use.

3. Significance:

Significance (Required)

This work has a number of contributions. Firstly, it demonstrates how to combine several complementary sequencing approaches to characterize in detail the transcriptional and translational state of a cell, as well as its overall growth rate to generate comprehensive expression-fitness maps. Secondly, it shows how the interwoven nature of cellular regulatory networks and the molecular interactions encoded within the genome can lead to cryptic responses in cellular behavior and fitness at a system-level that can only be understood by taking a detailed "bottom-up" approach. Finally, it suggests that some of these regulatory interactions may in fact "entrench" an organism's evolutionary path, by causing small genetic perturbations to propagate and potentially amplify their negative effect. While the results are compelling and well supported by experiments, the limited scope of the work makes it difficult to know whether this is in fact a rare or common occurrence. However, I do believe there is significance to these findings and that it will likely spur further studies to assess the generality of these findings.

Overall, I believe the work will have a wide appeal covering areas such as Systems Biology, Gene Regulation, Evolution, Quantitative Biology, Sequencing, High-throughput Technologies.

My field of expertise is in the quantitative measurement of core cellular processes (e.g. transcription and translation) using novel sequencing techniques and the application of

this knowledge to biological design. As such, I believe I have sufficient expertise to review this paper in detail.

Below is our point-by-point response:

 Reviewer #2 (Evidence, reproducibility and clarity (Required)):

The manuscript of Lalanne and coworkers address the cellular responses to varied translation termination factor expression in *Bacillus subtilis*. The authors set-up a system to fine-tune the expression of release factor RF1, RF2 as well as PrmC that post-translationally modifies RF1/RF2 to maximize their catalytic hydrolysis activity. They then monitor the fitness costs associated with overexpression or depletion of the factor by following the changes in growth rate. The set-up is nicely illustrated in Figure 1. The results in Figure 2 show that overexpression of RF1 and RF2 has relatively modest effect on the growth rate compared to overexpression of PrmC that leads to dramatic growth rate reduction.

By contrast, depletion of RF1 has a strong negative influence on fitness, whereas a similar level of depletion of RF2 had little influence on fitness.

PrmC overexpression appears to be correlated with the induction of the sigmaB regulon, however, the authors do not manage to ascertain why this is. By contrast, RF2 depletion also results in the induction of the sigmaB regulon and the authors demonstrate convincingly that this is due to a termination defect within the *rsbQ-rsbV* operon that contains an overlapping start-stop AUGA

A few points that the authors might consider discussing

1. The natural abundance of each RF in bacteria in relation to the usage of different stop codons in different organisms.

Response: We thank the reviewer for their suggestion. A correlation between RF abundance and stop codon usage across bacterial species has been previously reported (Korkmaz et al., 2014; Wei et al., 2016), which is corroborated by our quantification (see below). This correlation provides further evidence that the RF expression may be optimized to meet their demands in translation termination. We now include a new discussion in the main text (p. 9, lines 410-415):

"Our data thus corroborate several previous lines of evidence suggesting that RF expression might be precisely tuned. First, it was found that the relative expression between RF1 and RF2 correlates with stop codon usage between different species (Korkmaz et al., 2014; Wei et al., 2016). For instance, *B. subtilis* has a higher abundance of RF1 and more frequent UAG usage compared to *E. coli*, suggesting that RF1's expression setpoint meets translational demand (Methods)."

Below we include additional analyses that may be of interests to the Reviewer.

From our ribosome profiling quantification in *E. coli*, *B. subtilis*, *C. crescentus*, and *V. natriegens* (Lalanne et al., 2018), we can compare the relative usage of the three stop codons (frequency of stop codons weighted by expression) with abundances of RF1 and RF2:

Species	Release factor abundance		Stop codon usage		
	RF1 (prfA)	RF2 (prfB)	UAA	UAG	UGA
B. subtilis	0.085	0.034	0.888	0.064	0.049
E. coli	0.010	0.099	0.888	0.015	0.097
C. crescentus	0.022	0.033	0.594	0.225	0.180
V. natriegens	0.030	0.034	0.929	0.041	0.031

Table 1. Abundance RF1 and RF2 and respective stop codon usage in various bacteria. RF abundance (fraction of total protein synthesis rates estimated from ribosome profiling) for RF1 and RF2 in four bacteria. Stop codon usage refers to the fractional usage of each stop codon (weighted by synthesis rate of the upstream genes).

Despite the limited sample size, we find reasonable agreement with the expected correlation between codon usage and cognate RF abundance. In species with substantial differences between RF1 and RF2 abundances (*E. coli* and *B. subtilis*), the most heavily used non-UAA stop corresponds to the most highly expressed RF.

This argues in favor of expression tuning of these important enzymes and is consistent with the growth optimization we directly observe.

As a word of caution, although the low usage of UAG in *E. coli* and low expression of RF1 (reported long ago, e.g., (Adamski et al., 1994)) is well established, it should be noted that strain MG1655's RF2 factor harbors a debilitating missense A246T mutation near its active site (Dinçbas-Renqvist et al., 2000), which potentially complicates interpretation of the expression of *E. coli*'s release factors [interestingly, we do not see any difference in RF1 and RF2 expression from ribosome profiling data in strain NCM3722, which contains the RF2 variant without the A246T mutation (JBL, unpublished data)].

2. The role of the frameshifting mechanism in RF2 and how then RF1 levels are regulated.

Response: We thank the reviewer to raising the interesting topic of release factor expression regulation. We have added a section in our discussion to comment on RF2 regulation (p. 9, lines 415-420).

“Second, the gene encoding RF2 has a broadly conserved UGA-based frameshift event that autoregulates the expression based on its own activity (Baranov et al., 2002; Craigen and Caskey, 1986; Craigen et al., 1985). Interestingly, there are no reports of RF1 autoregulation to our knowledge, and we found that ectopic over- or under-expression does not affect its own promoter activity (Fig. S7). Therefore, a lack of autoregulation does not necessarily indicate that cells are less sensitive to small perturbations on its expression.”

The statement above includes an additional analysis on RF1 regulation that was motivated by the Reviewer's comment. In contrast to RF2, no definitive evidence exists on autoregulatory mechanisms for RF1. Following the Reviewer's comment, we realized that our dataset allowed us to search for evidence of endogenous regulation in *B. subtilis*: our RF1 expression strain has a markerless deletion of *prfA* and *prmC* genes, leaving the surrounding regions, and notably the promoter, intact. As such, possible unbeknownst regulatory mechanisms at the promoter level could be identified in our RNA-seq data under steady-state perturbation of RF1 levels. Quantifying the expression of the 5' untranslated region and operonic gene *ywkF* at the ablated *prfA* locus (presented in Fig. S7, reproduced below), we find no significant changes in expression across over 30-fold range in RF1 expression, arguing against such transcriptional regulatory mechanisms. Although this does not rule out other regulatory mechanisms at the post-transcriptional level, no such mechanisms have been documented for RF1 to our knowledge.

Fig. S7. Response of RF1 promoter to varying RF1 expression. (a) Schematic illustration of the genetic changes made to the endogenous *prfA* (RF1) locus in orthogonally tunable RF1/PrmC strain GLB438. The open reading frame for *prfA* is completely removed, leaving 5' UTR unperturbed (brown). The terminator intragenic to *prmC* is preserved (start codon of *prmC* removed, and in-frame stop added after the terminator). The intergenic and open reading frame for *ywkF* is unchanged. *ywkF* is solely transcribed from the *prfA* promoter. (b) Our steady-state expression series varying RF1 (experiment E2, Fig. 1a) and concomitant RNA-seq dataset allows us to assess whether there are transcriptional autoregulatory modules encoded around the endogenous *prfA* promoter region (with the caveat of the described perturbations). Shown is the quantification for the RNA level of the 5' UTR (brown) and *ywkF* gene (purple), showing no significant changes (two-sided t-test, $p > 0.2$) in expression across the full range of RF1 expression tested. Error bars correspond to standard deviation of bootstrap resampling estimates.

3. The authors observe queuing in front of the relevant stop codons upon RF depletion, however, do not discuss about readthrough events, which are usually competing with termination. Surprisingly, in this context the authors don't discuss the work from Mankin and coworkers showing sequestration of RFs from termination by peptides such as apidaecin leads to translational readthrough.

Response: We concur with the Reviewer about the importance of the recent work from Mankin et al. This paper was referenced in our original submission, but our literature management software improperly formatted its citation. The corrected reference to (Mangano et al., 2020) is now included in the revised manuscript.

Translational readthrough is indeed clearly visible in our ribosome profiling data from acute CRISPRi knockdown of RF1/PrmC and RF2. Using an approach analogous to Mangano and Florin et al, we quantified readthrough as the ribosome footprint density downstream of the stop codon (+5 to +45 bp) to the density in the gene body for isolated genes (no codirectional genes within 55 bp). We find five-fold increase in the median readthrough for genes that are terminated by the RF under perturbation (shown in a new panel in the main text, **Fig. 4b**, reproduced below). This new analysis is included in the section regarding translational phenotypes identified from ribosome profiling under RF depletion, **p. 7, lines 309-312**.

“The stop-codon-specific queuing is associated with translational readthrough downstream (Fig. 4b), consistent with a recent observation based on inhibition of peptide release by the antimicrobial apidaecin in *E. coli* (Mangano et al., 2020).”

Fig. 4b Translation readthrough phenotype under acute RF1/PrmC and RF2 CRISPRi knockdown. Translation readthrough score (following (Mangano et al., 2020)) is estimated as the read density downstream of the gene (from 5 bp to 45 bp after the stop codon) over the density in the body of the gene. Analysis is restricted to isolated genes (closest upstream and downstream genes on the same strand more than 55 bp) with a ribosome footprint read density >0.1. In both RF1/PrmC and RF2, nearly 5-fold change in translational readthrough is seen downstream of genes terminating with the stop codon cognate to the knocked-down RF (arrows). Points in beeswarm plot correspond to individual genes, with overlaid box plot highlighting the interquartile range (25th to 75th percentile, median red mark). Readthrough scores of 0 arise from genes with no reads downstream of the stop codon.

This additional analysis, in conjunction with (Mangano et al., 2020), also allows us to calibrate the depletion of RFs in our non steady-state CRISPRi perturbation. Given that apidaecin treatment (shown to lead to a nearly complete depletion of free RF in the cell) causes a >100-fold increase in readthrough, this suggests that our CRISPRi perturbation experiments only led to partial RF depletion at the moment of cell harvesting.

4. The efficiency of translation termination is well-known to be dependent on the context of the stop codon. Do the authors also observe such a trend. Especially, UGAC for RF2, one would expect to observe high levels of readthrough upon RF2 depletion.

Response: Further assessment of the sequence determinants that dictate susceptibility of certain genes and regulatory elements to RF perturbation is of great interest. We now include additional analyses for the effect of stop codon context on readthrough.

In our RF2 CRISPRi knockdown data, stratifying the translational readthrough (data from **Fig. 4b**) by stop codon and its next nucleotide, we observe only a modest ($\approx 2\times$, $p < 0.05$ two-sided t-test) exacerbation of the phenotype for the UGAC (and other non-U fourth nucleotides, shown to also have lower termination efficiency (Poole et al., 1995)) vs. UGAU tetranucleotide. The data is now shown in **Appendix Fig. 2**, reproduced below (we observe no such difference for the UAGN tetranucleotide under RF1/PrmC

knockdown, not shown). This relatively small difference is in line with the findings of (Mangano et al., 2020), which found limited influence of the nucleotide downstream of the stop codon on the accumulation of ribosome on stops following apidaecin treatment. This result is now included in our discussion of translational readthrough, p. 7, lines 312-314.

“We also observed a trend of tetranucleotide-dependent (UGAN) readthrough for RF2 knockdowns (Methods, Appendix Fig. 2) consistent with previous characterizations (Poole et al., 1995).”

Appendix Fig. 2. Translation readthrough phenotype under acute RF2 CRISPRi knockdown stratified by stop tetranucleotide.

Translation readthrough score (see Fig. 4b) for isolated genes (>0.1 ribosome footprint reads/nt) stratified by stop tetranucleotide for RF2 CRISPRi knock down. Points in beeswarm plot correspond to individual genes, with overlaid box plot highlighting the interquartile range (25th to 75th percentile, median red mark). Arrow points to the $\approx 2\times$ difference between UGAC and UGAU. * indicates $p < 0.05$ (two-sided t-test on log transformed values, excluding zeros).

As an additional point of interest, the importance of the 4th nucleotide in termination has not been studied outside of *E. coli*. Although indirect, one way to assess the influence of the 4th nucleotide is to determine the aggregated usage of each tetranucleotide stop signal by ribosome profiling. Interestingly, and as pointed out by the Reviewer, whereas *E. coli* (MG1655) displays a 16 \times increase in usage between the maximum UGAU (tetranucleotide usage 0.064) and minimum UGAC (tetranucleotide usage 0.004), no such difference is observed in *B. subtilis* (usage for UGAU and UGAC both at 0.015), suggesting that the immediate sequence context surrounding stop codons could have different consequences in different species.

Reviewer #2 (Significance (Required)):

Overall, the experiments are clearly performed and beautifully illustrated. Clearly, a lot of work has gone into this study but the end message that the cell regulates carefully RF concentrations is not surprising. Especially given that RF2 carefully regulates its own levels using an autoregulatory frameshifting mechanism. The major finding that the *rsbQ-rsbV* operon with the RF2 dependence leading to induction of the sigmaB regulon is in the end rather trivial since these regulators depend on RF2 for termination. Therefore, this manuscript is unlikely to have general interest to people in the translation field (such as myself) but rather those working in the field of synthetic biology.

Response: We thank the Reviewer for their positive assessment of our presentation and experimental methods, and for their judgment that our work will be of interest to synthetic biologists.

In our study, we used translation as a well-characterized system to interrogate the cellular response when enzyme concentrations are perturbed. Because the system is so well characterized, it allowed to ask whether the fitness effects are due to perturbations to the translation flux itself, or rather driven by spurious distal connections in the regulatory network. The end message we wish to convey is that enzyme expression is entrenched by spurious regulatory connections, suggesting that predictive bottom-up models of expression-fitness landscapes will require near-exhaustive characterization of parts.

Although our focus is on the cellular response, there are several interesting findings related to translation. First, we show that even though RF1 and PrmC are not subject to the strict autoregulation as RF2 is, cell

growth is similarly or even more sensitive to RF1 and PrmC abundance. Second, among the numerous regulators that depend on RF2 for termination, RbsV/RbsW is exceptionally sensitive to RF2 depletion (Fig. 4e). This result not only points to our incomplete understanding of translation regarding what makes this pair particularly susceptible, and further underscores the spurious nature of the cellular response to perturbations. We have expanded the discussions on the implication of these findings in the revised manuscript.

Reviewer #3 (Evidence, reproducibility and clarity (Required)):

In this paper, the authors use a combination of RNA sequencing, ribosome profiling and measurements of cellular composition and growth rate to gain insight into the multi-scale effects that perturbations to translation termination factors have on general physiological states and reproductive fitness using *Bacillus subtilis* as their model organism. Specifically, they find that perturbing the expression levels of peptide chain release factors in any direction has a negative effect on growth-rate. This negative effect was not due to a direct impact of the gene on the cell, but instead due to a chain of regulatory interactions that cause the activation of the general stress regulon. This leads to upregulation of a large chunk of the genome and an indirect impact on the expression of all other genes. Critically, the knock-on effects observed for the specific perturbations studied suggest that it may be difficult to predict expression-fitness landscapes of a cell, without carrying out a detailed mapping of all genes and the cell's physiological state.

Overall, the core findings in the paper are well justified by the data presented and the experiments appear to have been rigorously carried out.

Response: We thank the reviewer for their positive assessment.

My only concern is that it is unclear if biological replicates of the ribosome profiling were performed. Also, biological replicates are mentioned for the RNA-seq data, but no data is shown. Even a simple graph demonstrating the expression levels across these would be useful to be assured of no issues in reproducibility given the complex processing of the data involved.

Response: We now include additional analyses for biological replicates of RNA-seq and ribosome profiling experiments, which show the same high degree of reproducibility as we have demonstrated in previous studies (Johnson et al., 2020; Lalanne et al., 2018; Li et al., 2014).

With respect to RNA-seq quantification, we compared our 6 wild-type datasets (biological replicates except for different inert inducer concentrations, using the same batch of conditioned MCC medium) against each other in all possible pairs. The data is now included as **Appendix Fig. 1a** (referred to in the main text, **p. 4, line 138**), and is reproduced below. Across pairs, the mRNA level quantification shows a median FC_{10}^{90} (10th and 90th percentile in fold-change) between 0.86 to 1.16, and median R^2 of log-transformed data at 0.99. These statistics showcasing reproducibility of our RNA-seq methodology are now included in our description of our RNA-seq approach in the Methods, **p. S8, lines 313-320**.

(a) RNA-seq (mRNA level, rpkm)

Appendix Fig. 1a. Pairwise comparison of mRNA level quantification for biological replicates (wild-type). Each plot shows the mRNA level quantification (units of rpkm) for all possible pairs across our 6 wild-type RNA-seq datasets, with the only difference being different inducer [xylose and/or IPTG, genes *xyIA* and *xyIB*, endogenously responsive to xylose, are marked by +] concentration. Axes label correspond to the sample names (see Supplementary Data 3 for descriptions). All genes with more than 100 reads mapped in each sample are shown (number of genes indicated). The number of genes with >100 reads mapped, the R² for the log-transformed mRNAs and 10th to 90th percentile in fold-change FC₁₀⁹⁰ for each sample pair are shown.

Regarding ribosome profiling quantification, we now include comparisons between pairs of two replicates for wild type cells, and pairs of replicates wild-type with inert fluorescent protein expression, each pair of samples with their own batch of conditioned MCC medium. These samples were taken under different inducer concentrations, which are expected to affect the expression of two genes and not others. As indicated in **Appendix Fig. 1b** and reproduced below, the Pearson correlation of log-transformed footprint density is respectively of $R^2=0.98$ and 0.99 (genes with >100 reads mapped), with a 10th to 90th percentile of fold-changes between 0.83 to 1.17, and 0.91 to 1.12. These results are described in the Methods, p. S9, lines 339-345.

(b) ribosome profiling (ribosome footprint density, rpkm)

Appendix Fig. 1b. Reproducibility of ribosome profiling for biological replicates. Ribosome profiling data from wild-type (GLB115) and strain GLB455 (*B. subtilis* with ectopic fluorescent proteins under IPTG and xylose promoters). Each sample pair was grown in the same batch of MCC medium with different inducer concentration (which only affects the genes *xyIA* and *xyIB* [marked by +]). The number of genes with >100 reads mapped (n), the R^2 for the log-transformed rpkm, and 10th to 90th percentile in fold-change FC_{10}^{90} for each sample pair are shown.

Related to this, I see no mention of data availability in the paper. For this study to be useful to others, providing the raw data (unprocessed) would be essential (ideally in a public repository).

Response: We are sorry that the statement on data availability was buried in the original Methods section that was not a part of the merged PDF file. The raw sequencing data were submitted to Gene Expression Omnibus under the accession number GSE162169. The processed data, including fitness scores, mRNA levels, protein synthesis rates, were included as Supplementary Data Tables 1-9. We now moved the data availability statement to the main document at **p. 12, lines 512-516**.

The presentation of the work is excellent, with very clear figures and text that helped guide the reader through the results. There were a few minor comments:

1. Abstract: "in bacterium *Bacillus subtilis*" should read "in the bacterium *Bacillus subtilis*".

Response: This typo is now corrected.

2. Page 4: "found that under numerous ways" should read "found that under the numerous ways".

Response: This typo is now corrected.

3. The authors mention that changes in the expression level of RF1 impacted motility and biofilm genes, but not how this impacts fitness. Would they be able to experimentally identify origin of RF1 growth defects in the same way they did for PrmC? This is not essential for the main findings but would help strengthen the work.

Response: The cause of the growth defect under RF1 knockdown is indeed interesting. We now present evidence ruling out the hypothesis that the growth defect is caused by the expression decrease for motility and biofilm genes.

This hypothesis is driven by our result that ablation of SigB regulon rescues the fitness defect during PrmC overexpression (Fig. 3g) and by the observed downregulation of motility and *lyt* operons and upregulation of the *eps* operon during RF1 knockdown. To test this hypothesis, we used a strain without *sigD* (the motility sigma factor), which displays similar expression changes to what we observed in RF1 knockdown (Chai et al., 2009). Comparing the growth rates of wild-type to $\Delta sigD$, we found only a slight difference (<10%) in growth rates (doubling time 21.3 ± 0.8 min wild-type, 22.7 ± 1 min $\Delta sigD$). Because the difference is much smaller than the >30% growth defect measured upon RF1 knockdown, it appears that transcriptional changes to the motility regulon can only partially explain of the RF1 growth defect. These results are discussed on **p. 10, lines 459-463**. Further assessment will constitute interesting future research avenues.

4. It is difficult to know how generalisable the findings of this work are due to the very limited scope. It could be helpful for the authors in the discussion to consider and comment on how such approaches might be scaled-up to enable broader and more general studies of expression-fitness landscapes and where they will find most use.

Response: Indeed, the spurious nature of the expression-fitness landscape makes it difficult to generalize the exact mechanisms that we described here to other proteins. However, what is generalizable is our conclusion that such spurious connections limit the feasibility of bottom-up models for predicting fitness landscapes unless one has near-exhaustive characterization of all parts.

Our approach of mechanistic profiling of cell states under perturbations therefore provides a path forward that can be scaled up by recent developments in multiscale measurements. We now include a discussion for broader and more general studies on **p. 11, lines 473-480**.

“Various strategies can now generate expression-fitness landscapes for a large number of genes in parallel, for example using suites of promoters (Keren et al., 2016), genome-scale library of inducible gene expression (Arita et al., 2021), or tunable CRISPR perturbations (Hawkins et al., 2020; Jost et al., 2020; Mathis et al., 2021). Together with the advent of single-cell transcriptomics in bacteria (Blattman et al., 2020; Imdahl et al., 2020; Kuchina et al., 2020), these methods open the possibility of dissecting the molecular underpinnings of expression-fitness landscapes genome-wide, and to comprehensively identify instances of regulatory entrenchment.”

Reviewer #3 (Significance (Required)):

This work has a number of contributions. Firstly, it demonstrates how to combine several complementary sequencing approaches to characterize in detail the transcriptional and translational state of a cell, as well as its overall growth rate to generate comprehensive expression-fitness maps. Secondly, it shows how the interwoven nature of cellular regulatory networks and the molecular interactions encoded within the genome can lead to cryptic responses in cellular behavior and fitness at a system-level that can only be understood by taking a detailed "bottom-up" approach. Finally, it suggests that some of these regulatory interactions may in fact "entrench" an organism's evolutionary path, by causing small genetic perturbations to propagate and potentially amplify their negative effect. While the results are compelling and well supported by experiments, the limited scope of the work makes it difficult to know whether this is in fact a rare or common occurrence. However, I do believe there is significance to these findings and that it will likely spur further studies to assess the generality of these findings.

Overall, I believe the work will have a wide appeal covering areas such as Systems Biology, Gene Regulation, Evolution, Quantitative Biology, Sequencing, High-throughput Technologies.

Response: We thank the reviewer for their assessment that our work will be of appeal to a broad audience.

My field of expertise is in the quantitative measurement of core cellular processes (e.g. transcription and translation) using novel sequencing techniques and the application of this knowledge to biological design. As such, I believe I have sufficient expertise to review this paper in detail.

Response references

- Adamski, F.M., McCaughan, K.K., Jørgensen, F., Kurland, C.G., and Tate, W.P. (1994). The concentration of polypeptide chain release factors 1 and 2 at different growth rates of *Escherichia coli*. *J. Mol. Biol.* *238*, 302–308.
- Arita, Y., Kim, G., Li, Z., Friesen, H., Turco, G., Wang, R.Y., Climie, D., Usaj, M., Hotz, M., Stoops, E., et al. (2021). A genome-scale yeast library with inducible expression of individual genes. *BioRxiv* 2020.12.30.424776.
- Baranov, P. V, Gesteland, R.F., and Atkins, J.F. (2002). Release factor 2 frameshifting sites in different bacteria. *3*, 373–377.
- Blattman, S.B., Jiang, W., Oikonomou, P., and Tavazoie, S. (2020). Prokaryotic single-cell RNA sequencing by in situ combinatorial indexing. *Nat. Microbiol.* *5*, 1192–1201.
- Chai, Y., Normam, T., Kolter, R., and Losick, R. (2009). An epigenetic switch governing daughter cell separation in *Bacillus subtilis*. *Genes Dev.* *7824*, 754–765.
- Craigien, W.J., and Caskey, C.T. (1986). Expression of peptide chain release factor 2 requires high-efficiency frameshift. *Nature* *322*, 273–275.
- Craigien, W.J., Cook, R.G., Tate, W.P., and Caskey, C.T. (1985). Bacterial peptide chain release factors: Conserved primary structure and possible frameshift regulation of release factor 2. *Proc. Natl. Acad. Sci. U. S. A.* *82*, 3616–3620.
- Dinçbas-Renqvist, V., Engström, Å., Mora, L., Heurgué-Hamard, V., Buckingham, R., and Ehrenberg, M. (2000). A post-translational modification in the GGQ motif of RF2 from *Escherichia coli* stimulates termination of translation. *EMBO J.* *19*, 6900–6907.
- Hawkins, J.S., Silvis, M.R., Koo, B.M., Peters, J.M., Osadnik, H., Jost, M., Hearne, C.C., Weissman, J.S., Todor, H., and Gross, C.A. (2020). Mismatch-CRISPRi Reveals the Co-varying Expression-Fitness Relationships of Essential Genes in *Escherichia coli* and *Bacillus subtilis*. *Cell Syst.* *11*, 523-535.e9.
- Imdahl, F., Vafadarnejad, E., Homberger, C., Saliba, A.E., and Vogel, J. (2020). Single-cell RNA-sequencing reports growth-condition-specific global transcriptomes of individual bacteria. *Nat. Microbiol.* *5*, 1202–1206.
- Johnson, G.E., Lalanne, J.B., Peters, M.L., and Li, G.W. (2020). Functionally uncoupled transcription–translation in *Bacillus subtilis*. *Nature* *585*, 124–128.
- Jost, M., Santos, D.A., Saunders, R.A., Horlbeck, M.A., Hawkins, J.S., Scaria, S.M., Norman, T.M., Hussmann, J.A., Liem, C.R., Gross, C.A., et al. (2020). Titrating gene expression using libraries of systematically attenuated CRISPR guide RNAs. *Nat. Biotechnol.* *38*, 355–364.
- Keren, L., Haussler, J., Lotan-Pompan, M., Vainberg Slutskin, I., Alisar, H., Kaminski, S., Weinberger, A., Alon, U., Milo, R., and Segal, E. (2016). Massively Parallel Interrogation of the Effects of Gene Expression Levels on Fitness. *Cell* *166*, 1282-1294.e18.
- Korkmaz, G., Holm, M., Wiens, T., and Sanyal, S. (2014). Comprehensive analysis of stop codon usage in bacteria and its correlation with release factor abundance. *J. Biol. Chem.* *289*, 30334–30342.
- Kuchina, A., Brettner, L.M., Paleologu, L., Roco, C.M., Rosenberg, A.B., Carignano, A., Kibler, R., Hirano, M., William DePaolo, R., and Seelig, G. (2020). Microbial single-cell RNA sequencing by split-pool barcoding. *Science* (80-).
- Lalanne, J.B., Taggart, J.C., Guo, M.S., Herzel, L., Schieler, A., and Li, G.W. (2018). Evolutionary Convergence of Pathway-Specific Enzyme Expression Stoichiometry. *Cell* *749*–761.
- Li, G.W., Burkhardt, D., Gross, C., and Weissman, J.S. (2014). Quantifying absolute protein synthesis rates reveals principles underlying allocation of cellular resources. *Cell* *157*, 624–635.
- Mangano, K., Florin, T., Shao, X., Klepacki, D., Chelysheva, I., Ignatova, Z., Gao, Y., Mankin, A.S., and Vázquez-Laslop, N. (2020). Genome-wide effects of the antimicrobial peptide apidaecin on translation termination in bacteria. *Elife* *9*, 1–24.
- Mathis, A.D., Otto, R.M., and Reynolds, K.A. (2021). A simplified strategy for titrating gene expression reveals new relationships between genotype, environment, and bacterial growth. *Nucleic Acids Res.* *49*, e6.
- Poole, E.S., Brown, C.M., and Tate, W.P. (1995). The identity of the base following the stop codon determines the efficiency of in vivo translational termination in *Escherichia coli*. *EMBO J.* *14*, 151–158.
- Wei, Y., Wang, J., and Xia, X. (2016). Coevolution between Stop Codon Usage and Release Factors in Bacterial Species. *Mol. Biol. Evol.* *33*, 2357–2367.

RE: MSB-2021-10302, Spurious regulatory connections dictate the expression-fitness landscape of translation factors

Thank you for submitting your manuscript to Molecular Systems Biology. I have now read the revised manuscript and your point-by-point response to the comments of the Review Commons reviewers. I think that the performed revisions satisfactorily address the issues raised by the reviewers. As the study seems interesting and relevant for the audience of Molecular Systems Biology, I am glad to inform you that we can proceed with publishing it, pending some minor editorial issues listed below.

The authors performed the requested editorial changes.

Manuscript number: MSB-2021-10302R, Spurious regulatory connections dictate the expression-fitness landscape of translation factors

Thank you again for sending us your revised manuscript and for performing the requested minor edits. I am pleased to inform you that your paper has been accepted for publication.

Corresponding Author Name: Gene-Wei Li

Manuscript Number: MSB-2021-10302